

# The Trace Element Composition of Size Fractionated Suspended Particulate Matter Samples from the Qatari EEZ of the Arabian Gulf: The Role of Atmospheric Dust

[1*]Oguz Yigiterhan, [1]Ebrahim Mohd Al-Ansari, [2]Alex Nelson, [3]Mohamed Alaa Abdel-Moati, [2]Jesse Turner, [1]Hamood Abdulla Alsaadi, [2]Barbara Paul, [4]Ibrahim Abdullatif Al-Maslamani, [5]Mehsin Abdulla Al-Ansi Al-Yafei, and [2]James Wray Murray

[1]Environmental Science Center, Qatar University, P.O. Box 2713, Doha, State of Qatar

[2]School of Oceanography, University of Washington, Seattle WA, 98195-5351, USA

[3]Environmental Assessment Department, Ministry of Municipality and Environment, P.O. Box 39320,
Doha, State of Qatar

[4]Office of Vice President for Research and Graduate Studies, Qatar University, P.O. Box 2713,
Doha, State of Qatar

[5]Department of Biological and Environmental Sciences, Qatar University, P.O. Box 2713, Doha,
State of Qatar

*Correspondence to: Dr. Oguz Yigiterhan (oguz@qu.edu.qa)





**Abstract.** We analyzed net-tow samples of natural assemblages of plankton, and associated particulate matter, from the Exclusive Economic Zone (EEZ) of Qatar in the Arabian Gulf. Size fractioned suspended particles were collected using net-tows with mesh sizes of 50 μm (phytoplankton) and 200 μm (zooplankton) to examine the composition of plankton populations. Samples were collected in two different years (October 2012; April and October 2014) from 11 sites to examine temporal and spatial variabilities. We calculated the excess metal concentrations by correcting the bulk composition for inputs from atmospheric dust using aluminum (Al) as a lithogenic tracer and the metal/Al ratios for average Qatari dust. Atmospheric dust in Qatar is depleted in Al and enriched in calcium (Ca) relative to global average Upper Continental Crust (UCC) due to the geology of the outcropping sedimentary rocks and topsoil deposits in the source areas of the dust. To evaluate the fate of this carbonate fraction when dust particles enter seawater is uncertain, we leached a sub-set of dust samples using an acetic acid-hydroxylamine hydrochloride (HAc-HyHCl) procedure that should solubilize $CaCO_3$ minerals and associated elements. We found that Ca was removed and that the metal/aluminum (Me/Al) ratios for most elements increased after leaching because the change in sample mass resulting from the leach was more important than the loss of metals solubilized by the leach. Because surface seawater is supersaturated with respect to $CaCO_3$ and acid soluble Ca is abundant in the particulate matter, we only used unleached dust for the lithogenic correction. The concentrations of some elements in net-tow plankton samples appear to be mostly of lithogenic (dust) origin. These include Al, Fe, Cr, Co, Mn, Ni, Pb and Li. Several elements are mostly biogenic/anthropogenic origin. These include as Cd, Cu, Mo, Zn and Ca. The excess concentrations, relative to average dust, for most elements (except Cd) decreased with distance from shore, which may be due to differences in biology, currents, proximity to the coast or interannual processes.

**Key Words:** Particulate matter, Marine particles, Elemental composition, Trace metals, Plankton, Aeolian dust, Qatari dust, Acid digestion, Berger leaching, Qatar EEZ, Arabian Gulf, Persian Gulf

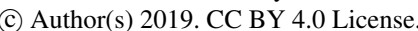


**1. Introduction**

      Data for the major and trace elemental composition of marine particulate matter are essential for our understanding biogeochemical cycling in the ocean. Identification of the biotic fraction is made complicated by the heterogeneity of particle assemblages. In most areas of the open ocean

marine particles are mostly composed of phytoplankton and zooplankton (e.g., Krishnaswami and Sarin, 1976; Collier and Edmond, 1984). Their composition is influenced by a variety of processes that include active uptake by phytoplankton, grazing by zooplankton, adsorption - desorption, particle aggregation and microbial remineralization (e.g., Turekian 1977; Bruland and Franks, 1983; Bruland et al., 1991; Alldredge and Jackson, 1995). But in the open ocean (e.g.,

Dammshauser et al., 2013; Rauschenberg and Twining, 2015; Ohnemus and Lam, 2015), and especially in marginal seas and coastal regions, lithogenic, abiotic particles are also important (Collier and Edmund, 1984; Twining et al., 2008; Iwamoto and Uematsu, 2014; Ho et al., 2007). These abiotic particles can include lithogenic minerals that are input from rivers, atmospheric dust, and sediment resuspension as well as authigenic particles such as Fe and Mn oxyhydroxides. The

relative contributions of biotic and abiotic particles can differ significantly. In some cases, the trace element contents of abiotic material can dominate the total composition (e.g., Ho et al., 2007).

      Distinguishing biogenic versus abiotic elemental composition is complicated. There have been few studies that tried to distinguish the relative contributions of biotic and abiotic particles in marine particulate matter. Biogenic and abiotic particles are difficult to separate physically using

most bulk sampling techniques. Characterizing the composition of biogenic particles is difficult due to the difficulties associated with sampling techniques and analyses (Martin and Knauer, 1973; Collier and Edmund, 1984; Ho et al., 2003; Twining et al., 2004; Rauschenberg and Twining, 2015). Bulk chemical analyses can be used to determine the relative contributions when the composition of the end-members are known. There can be significant differences in composition

spatially and among species of phytoplankton, making regional inferences difficult when considering only global average data sets. The most common approach has been to utilize metal/aluminum (Me/Al) and/or metal/phosphorus (Me/P) ratios. The biogenic portion can be determined after correcting for the lithogenic portion of the total concentration, using Al as the lithogenic tracer (Bruland et al., 1991; Ho et al., 2007; Yigiterhan et al., 2011; Yigiterhan et al.,

2018). Me/P ratios have been used as a way to normalize comparison of the biogenic portion of data sets from studies from different locations using different sampling techniques (e.g. Knauer and Martin, 1981; Kuss and Kremling, 1999). Ho et al. (2007) used a combined Me/Al and Me/P





approach to suggest that most of the trace metals associated with plankton, in samples from the coastal and open South China Sea, were associated with extracellular abiotic (inorganic) particles of atmospheric origin.

Particulate matter in seawater contains an important reservoir that can be solubilized and made biologically available. Some researchers have developed selective chemical leach approaches to identify labile trace metals. No chemical leach technique will be able to perfectly mimic natural processes but some have attempted to preferentially extract trace metals that are associated with intracellular material and particles adsorbed only to cells (e.g., Sañudo-Wilhelmy et al., 2001; Twining et al., 2011). The acetic acid-hydroxylamine (HAc-HyHCl) leach with a short

heating step, developed by Berger et al. (2008), was designed to identify the fraction of trace metals associated with cellular material and labile metals that may be adsorbed or associated with authigenic Mn/Fe oxyhydroxides (e.g., Murray and Brewer, 1977), that might be available to plankton.

There are also powerful techniques to analyze the composition of individual particles. The

element stoichiometries of individual cells can be determined using synchrotron-based X-ray fluorescence (SXRF), assuring accurate particulate measurements to ascertain biotic origin (Twining et al., 2003; Twining et al., 2004). This technique also provides an accurate comparison between species and cell types, along with an enhanced understanding of the distribution of elements within a cell - factors unresolvable when using conventional bulk sample analyses.

However, though surprisingly inexpensive, SXRF is time consuming and not always accessible, resulting in limited sample throughput.

All approaches to distinguish biotic and abiotic compositions have advantages and disadvantages. A systematic comparison of different methods was conducted by Rauschenberg and Twining (2015). They found that all bulk and leach methods included higher concentrations

of trace metals than those associated directly with cells, as determined using SXRF. They concluded that the Berger et al. (2008) leach was the best approach for quantifying biotic and associated labile particulate phases likely to be involved in marine biogeochemical cycling.

The area of our study was the Qatari Exclusive Economic Zone (EEZ), which is located at the center of the Arabian/Persian Gulf (here after "the Gulf"). This region is heavily impacted by

atmospheric dust, massive construction, heavy industrialization, oil and gas exploitation, and marine transportation, which renders it an ideal location to examine the influence of these coastal processes on particulate trace element compositions (Barlett 2004; Richer 2009; Al-Ansari et al.,





2015; Prakash et al., 2015; Yigiterhan et al., 2018). As there are essentially no rivers, atmospheric dust deposition is the main source of lithogenic particles to the surface marine waters of the Gulf. We examined the composition of local atmospheric dust in Qatar (Yigiterhan et al., 2018) and compared it with average upper continental crust (UCC; Rudnick and Gao, 2003) and local surface

terrestrial deposits (TSD, Yigiterhan et al., 2018) to determine the legitimacy of using each for a lithogenic correction. The atmospheric dust in Qatar is rich in Ca (as potentially soluble $CaCO_3$ and $CaSO_4$ minerals) and this complicates the use of Al as a tracer for the lithogenic correction. We studied the effect of distance from shore on the elemental concentrations in small and large size fractions of net tow particulate matter in coastal seawater samples, with the goal of

determining the influence of dust, plankton and anthropogenic sources in the Qatar EEZ.

The technique used in this study involved analyzing the total composition bulk samples collected using net tows of natural plankton assemblages. A subset of previously collected dust samples were leached using a modified version of the acetic acid-hydroxylamine hydrochloride procedure of Berger et al. (2008). The apparent excess concentrations (biogenic plus

anthropogenic) in the net-tow samples were determined by correcting for the lithogenic portion of the total concentration, using aluminum as the lithogenic tracer (Bruland et al., 1991; Ho et al., 2007; Yigiterhan et al., 2011) and the average composition of Qatari dust. This procedure provides an estimate of the non-lithogenic portion and includes labile elements that may be part of and available to plankton.

There have been few trace metal analyses carried out in coastal environments and marginal seas such as the Arabian Gulf (Bu-Olayan et al., 2001; Khudhair et al., 2015). This study was conducted to expand existing knowledge about the trace metal composition of particulate matter from biogenic and terrestrial sources in this coastal ocean region.

**2. Methods**

**2.1 Study region**

The Arabian/Persian Gulf is a 1,000 km long and 338 km wide arm of the Indian Ocean (Arabian Sea), covering 233,100-km$^2$ area. The basin is semi-enclosed and shallow with an

average depth of 36 m. It is one of the hottest, driest, most saline and dusty areas on the earth. In most of the Gulf seawater salinities range from 37 to 41. The Gulf is characterized by reverse-flow, estuarine circulation (in at the surface and out at depth) with no sill or saddle restriction at the southern entrance in the Strait of Hormuz (Reynolds, 1993; Swift and Bower, 2003).





The first comprehensive description of the region's hydrographic properties, nutrients and carbonate chemistry was presented by Brewer and Dyrssen (1985). They reported:

1. Strong south to north salinity gradients exist in the Gulf with salinities greater than $S = 40$ to the north.
2. Low nutrient levels everywhere in the surface waters except for the surface inflow from the Gulf of Oman through the Straits of Hormuz.
3. The ratio of carbon fixed as $CaCO_3$ to organic carbon fixed during biological production was about 2.5 to 1.

While there is a paucity of published data on primary productivity and diversity of phytoplankton in the region, existing information reveals north to south trends (Rao and Al-Yamani, 1998; Dorgham, 2013; Polikarpov et al., 2016) which are driven by particular environmental features (e.g., eutrophication, dust input and coastal (shallow) versus open (deep) water). Phytoplankton biomass (chlorophyll-a concentration), primary production, abundance, species diversity and species groupings, together with water quality parameters, were measured in the Qatari EEZ by Quigg et al. (2013). Of the 125 species identified, 70 % to 89 % were diatoms (mainly *Chaetoceros*), 8 % to 22 % were dinoflagellates and the remaining 2 % to 6 % were cryptophytes. No coccolithophorids were identified.

## 2.2 Sample collection

Particulate matter samples were obtained during three separate expeditions in the Exclusive Economic Zone (EEZ) of Qatar. The first expedition was conducted on the *R/V Janan* from 13 to 14 October 2012 and included eleven sampling sites in the Qatar EEZ, east of the Qatari Peninsula (Fig. 1). Sampling locations were selected to include samples from near-shore sites (Khor Al-Odaid, Mesaieed, Dukhan, Umm Bab) to off shore sites (Shrao's Island, Al-Edd Al-Sharqi (Ash Sharqi), Al-Edd Al-Gharbi, Halul Island, and High North), along with varying proximity to industrial and urban areas. Some sites, such as Al-Khor and Khor Al-Odaid are adjacent to environmentally protected areas in Qatar, while others, such as Mesaieed, Ras Laffan and Dukhan are subject to more industrial influence (Richer, 2009) and urbanization. Doha Bay is a major source of particles to the adjacent Gulf due to intense human activity in the city and associated suburbs. Dredging, effluent discharges, extensive construction activities, and heavy traffic are major sources of particulate loading to the bay (Yigiterhan et al., 2018).



The second expedition was conducted using speedboats on 01 to 02 April 2014. Sampling sites were located west of Dukhan, an oil producing region on the west coast (Dukhan Bay, Fig. 2a) and in shallow waters adjacent to the city of Doha on the east coast (Doha Bay, Fig. 2b). At each location, size fractioned, duplicate, net-tow samples were collected at three stations with

5       increasing distance from the coast along a relatively straight bearing. Sites were selected to investigate how industrialization and coastal processes influence near shore elemental concentrations on the western and eastern sides of the Qatar peninsula, where marine waters exhibit different physical and chemical properties.

**2.3 Sampling**

Samples were collected at all locations using custom designed plankton net-tows with mesh sizes of 50 µm and 200 µm to examine size fractionated particulate matter. We operationally defined samples collected with the 50 µm net as the small size fraction (phytoplankton) and those collected with the 200 µm net as the large size fraction (zooplankton). Both samples certainly

contained variable assemblages of phytoplankton, zooplankton and terrestrial particles of atmospheric origin (e.g., dust), however no attempt was made to visually determine their proportions. Metal free plankton ring nets of 200 cm length and 50 cm diameter (aspect ratio 4:1) were used to collect these samples. The particulate samples were rinsed into the cod-ends with surface seawater and transferred immediately, while onboard and under cover, into new, acid

cleaned 500 mL borosilicate glass jars with Teflon lids (Environmental Sampling Supply; Cutter et al., 2010). Each jar was carefully soaked for 2 days in 25 % $HNO_3$ and for another 2 days in 10 % HCL, and then rinsed 5 times with DI water. They were also rinsed 3 times with surface seawater from the same location before filling with samples from the cod-end. The glass jars were kept in a cooler under ice and transferred to 4° C refrigerators on board the vessel to allow particles to settle

in the dark. In the laboratory, within two hours of collection, the excess seawater was decanted and particulate samples were re-filtered under vacuum onto 142 mm diameter, 1 µm Nuclepore membrane filters. Particles were gently rinsed using double distilled deionized (DI) water to remove excess sea salt before transfer to acid cleaned vials for low speed centrifugation. The rinsing step was kept short to minimize breakup of fragile plankton (Collier and Edmond, 1984).

As a result, some sea salt still remained in the samples. The contribution of sea salt to dry weight and concentrations of Ca, Mg, K and Sr was corrected using concentrations of Na. Samples were




then frozen at -20º C and desiccated in a freeze drier for one week to 10 days before grinding and homogenizing at room temperature using well-cleaned agate mortar and pestle.

All sample sets were collected using nets towed horizontally at 1 to 2 meters depth at 1.5-knot speed for 10 minutes (~ 0.5 km distance). All equipment, techniques and protocols used for collection, sample handling, processing, drying and preparation were identical for both sampling periods. Special care was given to keep the nets at the surface in order to not collect any particles from the bottom, possibly influenced by resuspension.

Samples for atmospheric dust (n = 5) were collected from various locations around Qatar (Fig. 1, yellow dots on land), using automated and hand-sampling methods. These samples are a subset of the complete data set for Qatar dust analyses, which was presented in Yigiterhan et al. (2018). Each site was carefully selected to reflect differences in the influences of local traffic, construction, industry, or rural residency along the eastern coast of Qatar, where most of the Qatari population and industry are located. Dust accumulation is usually extensive, so collecting dust is very straightforward, but was conducted carefully to avoid contamination. Most individual dust samples were collected over 2 to 4 week intervals; however, when there was a dust storm, sampling intervals were as short as overnight. The sampling was done using passive dust traps (Siap+Micros brand). The particles were collected in an acid-cleaned glass bowl, which has automated protection from the rain with a digitally controlled closing-opening system, activated with rain sensors, for both dry and wet deposition. These traps were regularly visited during non-stormy autumn and winter seasons from September to February 2014. Sufficient dust particle accumulation (1 to 10 gm dry weight) was achieved after a sampling period of 1 month or more for some stations. When the targeted sample weight was achieved, samples were immediately collected. Sample extraction from the traps were mostly completed over a 48-hour period. Traps were visited after major (mega) storms to collect airborne transported particles during these events. The dust particles were trapped away from wind and anthropogenic local contaminants. This set of samples was analyzed in total (unleached) and leached forms.

### 2.4 Sample processing and analyses

Samples were digested in strong acid before analyses. Particulate samples were transferred into clean 50 mL plastic centrifuge tubes, and centrifuged for 15 minutes at ~3,000 RPM. Excess water was decanted at the end of the centrifuge cycle and the wet sample weight was recorded. The samples were then freeze-dried in the same 50 mL centrifuge tubes, and transferred to clean





tubes to be weighed again for dry weight. Samples were crushed and gently ground in the tubes to make a homogenous mixture. Weighed subsamples (250 ± 10 mg) were transferred to clean 50 mL trace metal certified plastic tubes. These samples were stored at room temperature in a desiccator until acid digestion.

Total acid digestion was completed on a heating block in acid cleaned Teflon tubes. Dried, powdered, homogenized samples (250 mg) were transferred into each tube, followed by addition of reagent grade (Ultra-Pure) 1mL HF, 5 mL $HNO_3$, and 2.5 mL HCl to each sample. The temperature of the hot plate was adjusted carefully and tubes were placed on the heating block in a fume hood with loose caps at 95°C for at least one hour. Addition of trace metal grade acids

(HCI and $HNO_3$) was repeated until a clear digest solution was obtained. Depending on the lipid content of the samples, 1 ml or more of trace metal grade oxidizer ($H_2O_2$) was added before further increasing temperature gradually to 135°C. Hot plate heating continued until near dryness. The sides of the tubes were rinsed consecutively with 1 mL $HNO_3$ and 5−10 mL of double distilled deionized (DDI) water. Digests were placed back onto the heating block in the fume hood with no

cap at 155°C, and boiled until clear. Solutions were transferred to clean, trace metal certified plastic bottles before rinsing the tube and cap at least 3 times with DDI water. The final sample volume was made up to 50 mL (by volume) by adding more DDI.  Samples were stored at room temperature until analysis by Inductively Coupled Argon Plasma Optical Emission Spectrometer (ICP-OES; PerkinElmer - Optima 7300 DV).

A subset of dust samples were also leached with acetic acid (HAc) and hydroxylamine hydrochloride (HyHCl) with a short heating step to remove the more labile particulate concentrations. Dried samples were weighed then treated with a modification of the leaching process described in Berger et al. (2008). Comparative analyses by Berger et al. (2008) and Rauschenberg and Twining (2015) showed that this technique most effectively digested the

biogenic portion, while leaving the more refractory elements intact. The leaching solution was composed of Ultra-Pure 25 % acetic acid (pH 2) with 0.02 M hydroxylamine hydrochloride as a reducing agent. 5 mL of leachate was added to 300 mg of homogenized, desiccated sample in a Nalgene test tube. For some zooplankton samples, less than 300 mg was available, in which case the total sample was used. The test tubes were placed in a beaker of water and raised to a

temperature of 95° C for 10 minutes, then left at room temperature for another 2 hours before centrifuging at 2,500 rpm for 5 minutes. After the leachate was removed from the sample and discarded, 5 mL of DDI water was added to the tube and shaken vigorously to ensure mixing.





Tubes were centrifuged at 2,500 rpm for 5 minutes, after which the water was removed. This rinsing step was performed three times and leachate was discarded. The leached and rinsed solid samples were freeze-dried, reweighed then totally digested using the procedure described above with the leachable fraction determined by difference. These samples were also analyzed by ICP-

OES (PerkinElmer - Optima 7300 DV) to determine the total elemental concentrations.

### 2.5 Precision and accuracy

Initial calibration verification (ICV 1640a) for ICP-OES was done before the sample analysis to meet the criteria for laboratory analytical accuracy. Duplicate samples, spiked samples, and

certified reference materials (PACS-2, PACS-3, MESS-3, DORM-4) were digested and analyzed twice with each batch of samples (typically 1 in every 10 samples). Three samples were analyzed in duplicate. In almost all cases, the average measured values were within the 95 % confidence limits of certified values, and thus the accuracy determined from this approach was comparable to or better than the precision. Limit of Detection (LoD) and Limit of Quantitation (LoQ) were also

calculated for each element. To ensure accurate measurements, four certified reference materials (CRM) were included in the analyses, i.e., PACS-2, PACS-3, MESS-3, and DORM-4. PACS-2 and PACS-3 are marine sediment reference materials for trace metals, collected in Esquimalt by the National Research Council of Canada (NRC) for the detection of trace metals. MESS-3 is also marine sediment collected in the Beaufort Sea by the NRC. DORM-4 is a reference material, based

on fish protein for trace metals, developed by the NRC. These CRM's were verified within 5 % of their expected values for all elements. All samples analysis were carried out triplicate and analyzed in ESC ISO 17025 accredited marine inorganic laboratory. The limit of detection was defined as three times the standard deviation of blank readings.

### 2.6 Sea salt (sodium) correction

The contribution of sea salt to the total mass and concentrations of Ca, Mg, K and Sr was corrected using Na. Of the major ions, Na is discriminated against by organisms and is therefore the best estimate of sea salt. The correction was done in Eq. (1) as follows:

$$Me_{corrected} = Me_{total} - Na_{sample} \text{ x } (Me/Na)_{seawater} \tag{1}$$

We used the Me/Na ratios in seawater from Pilson (2013).



However, interpreting the correction can be complicated. There is some uncertainty in this correction because of our assumption that all Na is from sea salt. The particulate matter in our study area is rich in carbonate grains, most likely originating as atmospheric dust. Recent marine carbonate deposits and skeletons have between 2,000 to 5,000 ppm or 0.2 to 0.5 % (Land and

Hoops, 1973; Milliman, 1974) and the average Na in our dust samples was 1.89 %. Thus, some of the Na in our marine particulate matter samples could be naturally in the samples, in the form of limestone and dolomite that originated from the aeolian dust. Other major sources of Na in detrital sedimentary rock types are detrital Na-feldspar and clay minerals. For example, the average Na in our uncorrected particulate samples was 60,000 ppm, which equals 6 %. We could argue that only

90 to 95 % of the Na in our samples is due to sea salt. But, because the Na varies so much from sample to sample, we decided to use the total Na in each sample for the sea salt correction. If we overestimate the sea salt correction our sample mass would be too low and the metal concentrations would be too high. However, the magnitude of this uncertainty is within the natural variability, so it can be ignored.

## 2.7 Lithogenic correction

The total digestion procedure dissolved all abiotic as well as biogenic material in the samples. In order to determine the biogenic portion of the metal concentrations, we used aluminum (Al) as a lithogenic tracer and assumed that all Al in the samples was of lithogenic origin. Aluminum is

an effective tracer for lithogenic or terrigenous inputs due to its high crustal abundance and generally little or no biological uptake (Ho et al., 2007). While Al is considered to be generally insoluble (Hodge et al., 1978), its small solubility (1.5 % to 5 %) has been used to model atmospheric dust deposition rates (Measures and Brown, 1996; Measures and Vink, 2000). Titanium has been proposed to be a better tracer than Al for crustal materials (Ohnemus and Lam,

2015). Unfortunately, Ti was not included in our analyses so that option was not available for us. Because river input is absent in the Qatari EEZ, and the underlying sediments are mostly carbonates, we assume that scavenging and biological cycling do not apply for our atmospheric dust and water column samples. Because the lithogenic samples can consist of materials from different origins, it is best to compare analytical data using Me/Al ratios, rather than absolute metal

concentrations to correct for dust contributions (Yigiterhan and Murray, 2008).

The Me/Al ratios were calculated for two possible lithogenic materials by dividing the concentration of the desired metal by the Al concentration. For determining lithogenic





contributions to our samples, we utilized average total (unleached) Qatari dust (Table 1 in
Yigiterhan et al., 2018) and average leached Qatari dust (Table 1, this study). We multiplied these
ratios by the Al concentration in each sample, to give a value for the lithogenic contribution of
each element. After subtracting this value from the total concentrations, the remaining

concentration was assumed to be the excess metal (Me$_{excess}$) of biogenic, authigenic or
anthropogenic origin (Yigiterhan and Murray 2008; Yigiterhan et al., 2011). Excess metal
concentrations were defined in Eq. (2) as:

$$Me_{excess} = Me_{total} - Al_{total} \text{ x } Me/Al_{dust} \tag{2}$$

### 3. Results

#### 3.1 Dust

Five dust samples, previously collected, were included in this study. They were analyzed
before and after leaching, using the Berger et al. (2008) leaching procedure. The elemental
concentration data for the total (unleached) and leached dust samples are reported in Table 1. The
goal of this table is to put some limits on the composition of lithogenic particles, originating from
terrestrial rocks that are input to the Qatari EEZ from the atmosphere. The total and leached
compositions are compared with global average Upper Continental Crust (UCC) from Rudnick
and Gao (2003) as ratios of total / UCC (column 4) and leached / UCC (column 7). We show this
comparison because UCC is sometimes used for the lithogenic composition when local data sets
are not available and we want to show how Qatari dust differs from UCC. Ratios greater than 1
indicate that the concentrations of elements in these dust samples from Qatar are greater than UCC
values. These elements include Ca, Cd, Li, Mg, Mo, Sr and Zn. As reported previously (Yigiterhan
et al., 2018), Qatari dust samples had Ca concentrations 5.3 times larger than in UCC. Sr (2.4x) >
Mg (1.9x) ≥ Cd (1.29x) > Li (1.8x), were also enriched relative to UCC. Ratios less than 1 suggest
that Qatari dust is depleted relative to UCC. The ratios of leached to total compositions are also
included in Table 1 to show the impact of leaching (column 6). The elements removed by the leach
have a ratio <1 and include Ca and Li. All other elements may also be removed but have higher
concentrations in the leached material because of the reduction of mass (loss of CaCO$_3$). The total
and leached values are also compared using their Me/Al ratios, which is one approach often used
for comparing samples with variable compositions (columns 8, 9, 10). Clearly, use of UCC





concentrations for the end-member composition of atmospheric dust would not be appropriate for our study because Qatari dust has large carbonate mineral content reflecting the composition of the calcareous rocks and thin top soils in the source regions of Qatar.

**3.2 Particulate samples**

The elemental composition of the two size classes of particulate net-tow samples are given in Tables 2 (2012) and Table 3 (2014). The raw data (column R) for the sea-salt corrected composition of the size fractionated marine particulate matter samples from the 2012 and 2014 cruises are presented by net tow mesh size, thereby representing the small and large size
fractionated particulate fractions. We calculated the lithogenic contributions for each element (column L) using average total (unleached) Qatari Dust (from Table 1) and the resulting excess concentrations for each element (column E). Al only has column R as it was used to calculate the lithogenic fraction. Average values and standard deviations for each size fraction are shown in the last row of each section. The cumulative average and standard deviation are shown at the bottom.
We also calculated the excess concentrations assuming that the lithogenic fraction had the average values of leached Qatari dust (from Table 1). The excess concentrations calculated that way were often negative because the "apparent" lithogenic correction was so much larger. As these values are unrealistic, those calculations are shown in Supplemental Material Table S1 (2012) and Supplemental Material Table S2 (2014).
Unfortunately, neither Ca nor P analyses were included in this data set. This this deficiency will be corrected in future efforts. The present data set is still valuable as it represents the first with concentrations of particulate trace elements in the Qatari EEZ.

**4. Discussion**

Most marine particulate matter is composed of material of biotic origin with minor contributions from atmospheric dust. The Qatari EEZ is an especially unique site to study marine particulate matter because of the large deposition rates of atmospheric dust. Our goal was to use the composition of local atmospheric dust to correct for the contribution of dust to size-fractionated marine particulate matter. Then we ask: are the remaining concentrations consistent with previous
studies of bulk plankton and can anthropogenic signals be identified?



### 4.1 Regional dust composition

Qatar is in a region heavily affected by dust storms originating from a multitude of sources. Back trajectories show that winds come from a variety of locations, but mostly either from the northwest (northern Saudi Arabia and Iraq) or the southeast (Oman) (Yigiterhan et al, 2018).

Because dust is potentially such an important source of lithogenic particles to the surface waters of the central part of the Arabian Gulf, determining the composition of the total and potentially soluble dust end-member concentrations was an important part of this study. Ideally, with well-established end-members, we could analyze the composition of the size fractionated particulate matter and determine the relative contributions of the dust and plankton end-members and possible

anthropogenic contributions. In the absence of knowledge about local sources, researchers often make the lithogenic correction using the composition of average Upper Continental Crust (UCC). Fortunately, in this study, the composition of local duct in Qatar has been well characterized. The composition of Qatari dust is carbonate rich which is different from the composition of atmospheric dust from most other regions around the world, which is dominated by aluminosilicate

material (e.g., Prospero et al., 1981; Mahowald et al., 2005; Lawrence and Neff, 2009; Muhs et al., 2014; Patey et al., 2015).

The concentrations of all major and trace elements in Qatari dust are different from those in average upper continental crust (UCC), (Table 1). The most notable differences between the two are the depletion of aluminum and the enrichment of calcium in Qatari dust. Yigiterhan et al (2018)

showed that the average composition of Qatari dust has concentrations of Ca much higher than those predicted by the crustal Ca/Al ratios, suggesting that $CaCO_3(s)$, and possibly $CaSO_4(s)$ minerals are a major component of the total mass. In UCC, aluminum has a higher concentration (8.2 %) than calcium (2.6 %). However, in unleached Qatari dust, the two elements switch positions, with aluminum decreasing to 2.1 % by weight versus calcium, which is elevated to 13.7

% by weight (Table 1). Qatari dust has over 4 times the amount of calcium, and about a quarter of the amount of aluminum compared to UCC. By converting Ca to $CaCO_3$ and comparing it with the total mass, we calculate that carbonate minerals make up 35 % or more of the total mass. The reason for this is due to the abundance of carbonate minerals in rock outcrops and soils in the Qatar peninsula, thereby allowing carbonate minerals (calcite and dolomite) to be a major component of

dust via erosional processes. The majority of the Qatari Peninsula is underlain by uniform limestone beds of Miocene and Eocene age (ESC, 2014) and the source areas to the north and



south are similar. As a result, the composition of Qatari dust from northern and southern regions is very similar for most elements (Yigiterhan et al., 2018).

The enrichment of calcium carbonate, along with other possible metal-$CaCO_3$ associations, leads to uncertainty regarding how to best use the dust samples for the lithogenic correction. In order to obtain an elemental composition for dust with carbonate minerals removed, and to understand the possible impact of solubilization, a subset of samples of Qatari dust were leached with the leach procedure of Berger et al., (2008). Calcium carbonate is more soluble than aluminosilicates. Samples leached with HAc:HyHCl may in fact be a more accurate representation of the actual concentration to use for the lithogenic correction. The compositions of Qatari dust before and after the leaching process are compared in Table 1. Because the emphasis of study was on particles, we focused on how the composition of the particles change rather than on how much of the different elements was solubilized. Certain elements that are usually associated with calcium carbonates (e.g., Mg, Sr, Li) are elevated relative to UCC in dust samples prior to leaching. If elements associated with these carbonate minerals are solubilized in seawater, our results show that their concentrations in the leached dust samples could be elevated relative to aluminum, causing the Me/Al ratio to be elevated as well. You would normally expect that this leach would remove and lower the concentrations of reactive elements. However, all elements other than Ca and Li increased in concentration because the mass removed by the Berger leach was more important than the removal of the individual elements by the leach. As a result of mass removal during the leach, the concentrations of Mo (3,350 %), Ni (511 %), As (234 %), Ba (204 %), Cr (179 %), Pb (148 %), Zn (149 %), Fe (132 %), K (111 %), and Al (105 %) in the leached solids more than doubled. The difference between total and leached Qatari dust samples expressed as the percent difference from the total composition is shown as a bar graph in Fig. 3. One important outcome of the leach experiment was that we showed that the high Ca content could be solubilized. We did the lithogenic corrections using leached dust and this made the corrections so large that, in most cases, the corrections were unrealistic and larger than the initial concentrations so those results are not shown.

## 4.2 Fate of dust in seawater

It is uncertain how the composition of Qatari dust might be modified when the particles enter surface seawater. Do the particles keep the composition they had in the atmosphere or does the $CaCO_3$ dissolve (completely or partially), thus changing the Me/Al ratios of the residual material?





Carbonate mineral dissolution would be driven by the saturation state in the water column. Very little is known about the carbonate chemistry in the Arabian Gulf, though a limited early study in 1977 (Brewer and Dyrssen, 1985) suggested that the surface seawaters were saturated with respect to calcite and that $CaCO_3$ is being formed. One possible benefit of this carbonate rich dust might

be to buffer future increases in ocean acidification (Doney et al., 2009) in the Arabian Gulf.

As a result of this weak acid leach much of the carbonate mineral (and associated element) fraction in Qatari dust was removed into leachate solution. Equating the conditions of the Berger et al. (2008) leach with what happens to dust particles upon deposition in seawater is highly problematic. However, the leached dust gives the most extreme outcome where most of the

$CaCO_3(s)$ and associated elements were solubilized and removed. The elements that have net removal by the leach have a ratio $< 1$ and include Ca and Li. The concentrations of all other elements increased in the leached material because of the reduction of mass. The ratio for Na was close to 1.0 (0.97), which suggests that our sample rinse procedure after collection removed most of the sea salt contamination. The data in Table 1 show that after the leach Ca decreased by 45 %

and Li by 44 %.

Unfortunately, Ca was not included in the particulate matter data set presented here, but we have recently found high concentrations of particulate Ca in a new set of particulate net-tow samples (Yigiterhan, unpublished data). Significant Ca ($> 95$ %) is removed from both size fractions by the HAc-HyHCl leach suggesting that $CaCO_3(s)$ is present in those water column

particulate matter samples. $CaCO_3$ forming plankton are rare in the Arabian Gulf (Quigg et al., 2013; Polikarpov et al., 2016). Thus, these data suggest that $CaCO_3$ in Qatari dust does not dissolve upon entering seawater after atmospheric deposition. In this study, we only used the unleached dust composition for the lithogenic correction in Tables 2 and 3.

**4.3 How much of the trace element composition is controlled by dust?**

Atmospheric dust makes a major contribution to particles in the water column of the Arabian Gulf. After making the lithogenic correction do any elements have excess concentrations that can be attributed to plankton or anthropogenic sources?

To correct for the lithogenic contribution to the plankton samples, we used aluminum as a

tracer, assuming that all Al in each sample came from dust of terrestrial origin in the Arabian Gulf region. We used metal to Al ratios for average local (unleached) Qatari dust (Table 1), to subtract the lithogenic contribution for each element and obtain the excess metal (non-lithogenic)




concentrations in the samples (column E for each element in Tables 2 and 3). For each element, column R gives the original composition. Column L gives the lithogenic correction using the Me/Al ratio average Qatari dust (in parentheses). Column E gives the excess metal. For some elements in some samples, the dust correction (L) was larger than the original signal (R) and this

resulted in negative values for excess metal (E) concentrations. As negative values are impossible it means that, in some cases, the lithogenic correction, done with total (unleached) Qatari dust data overcorrected for the lithogenic input. We treated this as random error and argue that it is best to focus on the rows with the average and standard deviation for the cumulative data set for each element. The data are grouped by filter size with the 50 μm samples at the top and the 200 μm

samples at the bottom,

The difference between the average excess metal concentrations in small (phytoplankton) and large (zooplankton) net-tow samples are shown in Fig. 4. The average compositions of the two size classes were in good agreement when you consider the standard deviations. We ran t-tests to determine if the average small and large plankton samples were statistically different. For the 2012

data, there were only a small statistical differences for Ni and V. For the 2014 data, there were small statistical differences for Cr, Cu, Fe and Ni. Thus, we combined the data sets and report the grand average and standard deviation for all samples as the cumulative rows at the bottom.

We conducted statistical tests to determine if the cumulative means of the excess concentrations (E) were different from zero. If column E was not statistically different from zero

it would support the argument that the elemental concentrations in the particulate samples were totally due to the presence of lithogenic material (Qatari dust) and that contributions from plankton and anthropogenic sources would be insignificant. The outcome of the statistical evaluations are shown in Table 4 (for 2012 data) and Table 5 (for 2014 data).

All tests to determine if E was different from zero were one-sample tests using the combined

plankton for a given year.  They were also one-sided with the alternative hypothesis that the sample was greater than zero. To decide which test to use, we first had to use the Shapiro test to establish if the sample concentrations were normally distributed. If the Shapiro test is greater than 0.05, distributions are normal. The outcomes regarding normality are listed in the 6th column. Most distributions are not normal. If distributions are normal, you can use the t-test. If not normal, you

have to use the Wilcoxon-Rank Test. The 7th column shows the test used. For the 2012 sample set, only Ni was normally distributed. For 2014 As, Cd, Fe and Ni were normal.




We used the Wilcoxon-Rank test with alpha = 0.5 and the output is a p value for the difference of the median from zero. Those p values are shown in the 4th column. The answer to the question "Is the value of column E different from zero?" is given in the 3rd column.

For 2012 only two elements (Ba and Fe) was not different from zero. This means that the input of Qatari dust can explain the Ba and Fe composition in Qatari marine particulate matter. The rest were different from zero. There is a statistically recognizable biological/anthropogenic component. For 2014, Ba, Cr and Mo were not different from zero. Traditional box plots showing these differences visually are shown in Fig. 5.

## 4.4 Metal to Aluminum ratios

Another approach for examining the question regarding importance of input from Qatari dust and excess metal concentrations is to use plots of metal versus aluminum concentrations (Fig. 6). As Al has no biological function and is input only from atmospheric dust, we argue that those elements that correlate strongly with Al and have the same Me/Al ratio as dust, are controlled by dust input. The Me/Al plots include a best fit linear regression (solid line) and lines representing the Mw/AL in average upper continental crust (UCC)(long dashes)(Rudnick and Gao, 2003) and average Qatari dust (short dashes)(Table 1).

The grouping of elements is pretty clear. The elements with trends parallel to Qatari dust include Co, Cr, Fe, Mn, Ni, Pb and Li. V increases with Al but at a higher rate. This may be due to scavenging of V from seawater. Ba increases with Al but at a lower rate. Perhaps Ba can be solubilized out of the dust when it enters seawater. Mo mostly increases with Al in agreement with Qatari dust with the exception of 6 samples, from the 2012 data set, at low Al that have large excess Mo. As and Cu are uniformly higher than expected for Qatari dust and Cd is higher, especially at low Al concentrations. It is tempting to speculate that these elements are higher due to biological enrichment. Me/P ratios are required for comparison with other studies of plankton composition. But in the absence of data for P, we cannot separate plankton from anthropogenic sources in this study

## 4.5 Plankton and anthropogenic sources

In addition to Qatari dust, plankton and anthropogenic sources are likely contributors to particle chemistry. For those elements where the average and standard deviations for the cumulative average particulate samples are not significantly different from zero, it means we





cannot identify the plankton composition by difference. That was the case for Ba and Fe in the 2012 samples and for Ba, Cr and Mo in the 2014 data. For those elements, there are no statistically significantly excess concentrations in our data set. This is an oligotrophic region with typical chlorophyll concentrations of 1.0–1.5 µg l$^{-1}$. It appears that plankton are just not abundant enough

relative to the input of dust to be detected by the uncertainties in this approach.

For each element, we calculated the percent of the total concentration that was represented by the difference of the total minus dust (E). The percentages are given in Table 6. Based on the combination of the statistical analysis and the plots of Me versus Al we propose the following generalizations. These are proposed here as hypotheses that can be tested with future data sets that

are more complete with Ca and P analyses.

Our tentative generalizations are that:

1. Some elements in net-tow plankton samples are mostly of lithogenic (dust) origin
   (E < 50 % of total). These include Al, Fe, Cr, Co, Mn, Ni, Pb and Li.

2. Some elements are mostly biogenic/anthropogenic origin (E > 80 % of total).

These include As, Cd, Cu, Mo, Zn and Ca.

### 4.6 Regional trends

The excess metal concentrations are what can originate from plankton and anthropogenic sources. In order to examine regional variability we plotted the excess metal concentrations from

Tables 2 and 3 versus distance from shore (Fig. 7). Best fir linear regressions are included to represent trend lines. The results are striking.  Excess concentrations for most elements decreased with distance from shore (Fig. 7). Ten of the thirteen elements (As, Co, Cr, Cu, Fe, Mn, Mo, Ni, Pb, V, and Zn) had high concentrations in the 0–10 km range. Several elements may also show a decreasing trend with distance greater than 10km (Co, Cu, Fe, Mn, Ni, Pb but the statistics are

poor. Cadmium was the only element for which there was a significant increasing trend with distance, with an r-squared value of 0.68. As there was no increase in chlorophyll in the 0–10 km range it is unlikely that the high excess metal concentrations were due to biological processes. Future studies should examine this more closely.

We hypothesize that abiological coastal processes are the most likely cause of these gradients

but the exact nature of the processes is unclear. As a country heavily affected by rapid population growth and new construction, near shore sampling sites could be influenced by pollutants from disposal from outfalls (surface and groundwater) or industrialization. Near shore dredging and





reclamation are also important processes used to recontour the coastline of Qatar and to obtain $CaCO_3$ and silicate minerals for cement factories. It is unlikely that these high values are due to dust deposition as the excess concentrations are calculated relative to average Qatari dust. This nearshore enrichment phenomenon emphasizes the need for more detailed sampling of plankton

and anthropogenic sources in regions close to shore.

## 5. Conclusions

Net-tow samples of natural assemblages of plankton from the Exclusive Economic Zone of

Qatar and the Arabian Gulf were analyzed for elemental composition. Samples were collected using net tows with mesh sizes of 50 μm (phytoplankton) and 200 μm (zooplankton) to examine size fractionated plankton populations. Samples were collected in two different years (2012 and 2014) to examine temporal variability. Atmospheric dust from Qatar is the main source of lithogenic particles to surface seawater. Qatari dust is depleted in aluminum and enriched in

calcium relative to the global average upper continental crust (UCC) values. The fate of the carbonate fraction when dust particles enter seawater is uncertain, thus, we leached a sub-set of dust and plankton samples with an acetic acid-hydroxylamine hydrochloride procedure to solubilize $CaCO_3$ minerals and associated elements. We found that a significant amount of the Ca was solubilized but that the metal/Al ratios for many elements increased after leaching because the

change in mass due to removal of $CaCO_3$ was more important than the loss of metals solubilized by the leach itself. Because the surface seawater of the Arabian Gulf appears to be supersaturated with respect to $CaCO_3$ and the concentrations of particulate Ca are large, we assumed that the $CaCO_3$ in dust does not dissolved on entering the seawater.

Excess elemental concentrations due to plankton and anthropogenic inputs were obtained by

correcting total concentrations for input of atmospheric dust using aluminum as a tracer and the average metal/Al ratios for average Qatari dust. We calculated the lithogenic correction using unleached dust compositions. Statistical analysis showed that for some elements the excess concentrations were indistinguishable from zero. This included Ba and Fe in 2012 and Ba, Cr and Mo in 2014. This suggests that these elements the concentrations in particulate matter can be

explained as having only a dust origin. For a second approach we examined how the Me/Al ratios compared with the ratios in average Qatari dust. The elements with trends parallel to Qatari dust



included Co, Cr, Fe, Mn, Ni, Pb and Li. V increases with Al but at a higher rate. Our tentative generalizations based on the statistical and Me/Al ratio approaches are that:

1. Some elements in net-tow plankton samples are mostly of lithogenic (dust) origin (E < 50 % of total). These include Al, Fe, Cr, Co, Mn, Ni, Pb and Li.

5    2. Some elements are mostly biogenic/anthropogenic origin (E > 80 % of total). These include As, Cd, Cu, Mo, Zn and Ca.

The excess concentrations relative to average Qatari dust for most elements were higher in the 0 to 10 km distance from shore. Cd was the one exception and had a statistically significant increase with distance from shore. It was not clear if variability was due to differences in biology

10    or non-biological (possibly anthropogenic) processes.





**Figure captions**

**Figure 1:** Plankton sampling locations (red dots) for the two offshore sampling cruises in 2012 and 2014. The near shore samples are shown separately in Fig. 2. The land based sampling stations

5      for atmospheric dust are shown with yellow dots. Ocean Data View (ODV) was used for plotting the figure (Schlitzer R., 2017).

**Figure 2:** Station locations for near shore sampling near Dukhan Bay and Doha Bay. ODV was used for plotting the figures (Schlitzer R., 2017).

**Figure 3**:  The difference between unleached and leached Qatari dust samples expressed as the percent difference from the unleached composition**.** Molybdenum (Mo) is not shown because its difference (+3,350%) was off scale.

15    **Figure 4:** The difference between the excess metal concentrations in the small size fraction (50 µm, phytoplankton) and large size fraction (200 µm, zooplankton) data sets. These represent the average of the cumulative data in the two data sets for 2012 and 2014. The vertical axis is logarithmic. The value for Ba in the phytoplankton data is not shown because it was negative due to overcorrection and could not be plotted on a logarithmic graph.

**Figure 5:**  Box plots for statistical texts. Traditional boxplot where the whiskers extend to 1.5 times in the interquartile range and then any points that fall outside of that are shown as individual dots on the graph. The dashed line at 0 serves as an indicator of where the distribution lies relative to 0.

**Figure 6:** Metal/Al ratios for the 2012 and 2014 net-tow samples. The dashed lines represent the Me/Al in average Qatari dust (Yigiterhan et al., 2018) and Upper Continental Crust (Rudnick and Gao, 2003) values. The solid line represent to best-fit line for cumulative values of plankton samples. All concentrations are given in ppm.

**Figure 7:** The excess concentrations of each element in alphabetical order after lithogenic correction using average Qatari dust versus the shortest distance from the sampling site to land. The graphs show the combined data from four data sets: 50 µm and 200 µm mesh net tows (called phytoplankton and zooplankton, respectively) for both 2012 and 2014.





**Figures**

**Figure 1:**

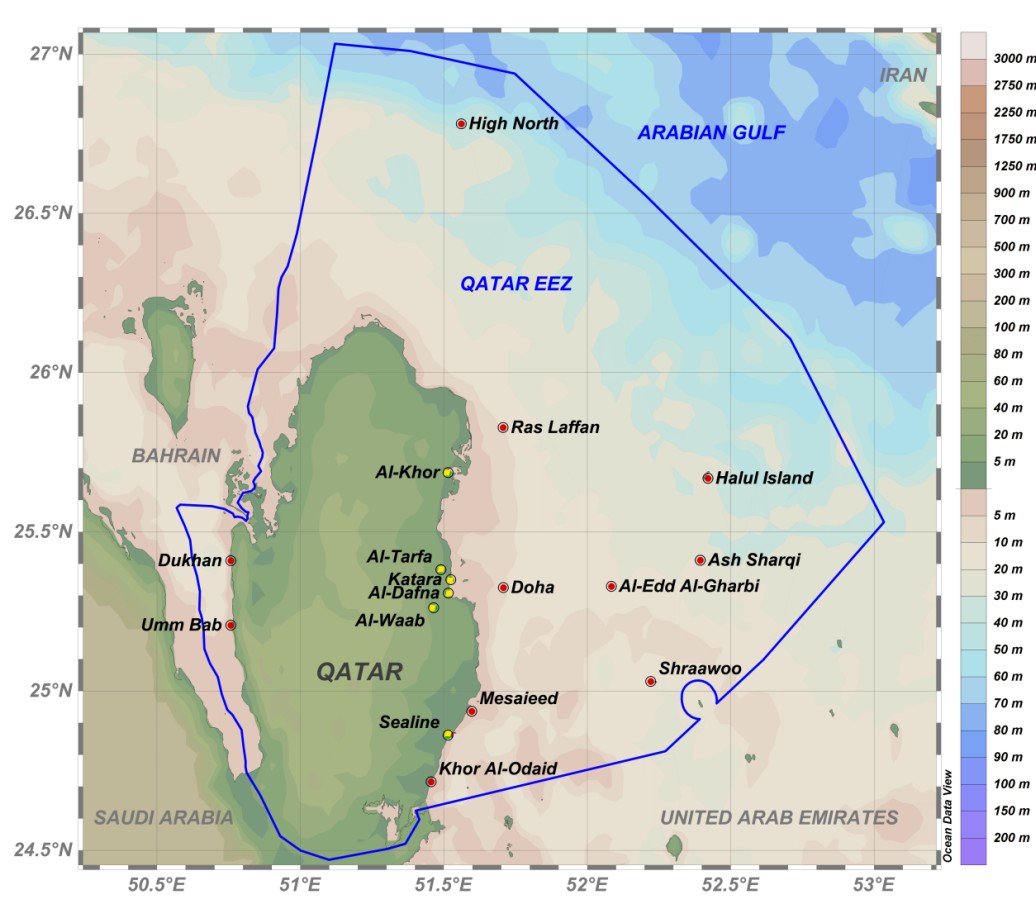



**Figure 2:**

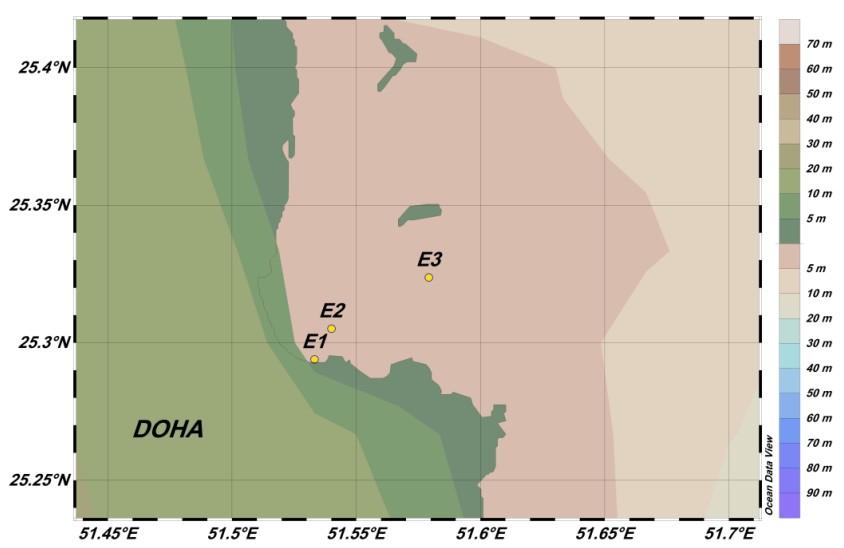

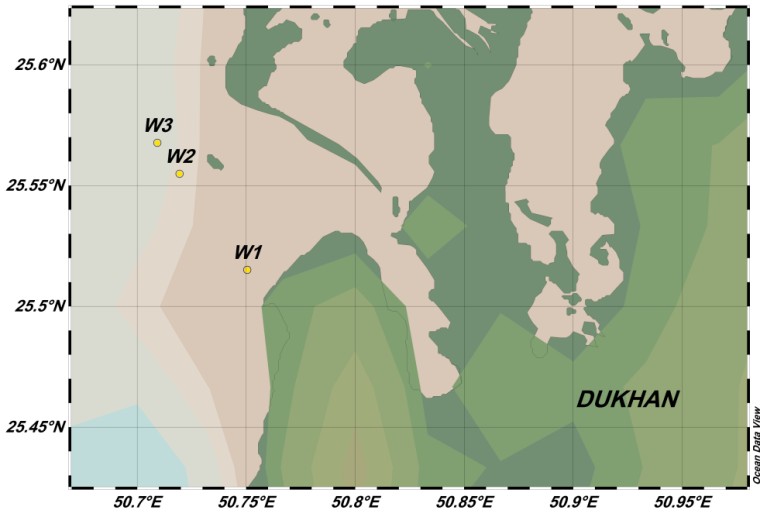



**Figure 3**:

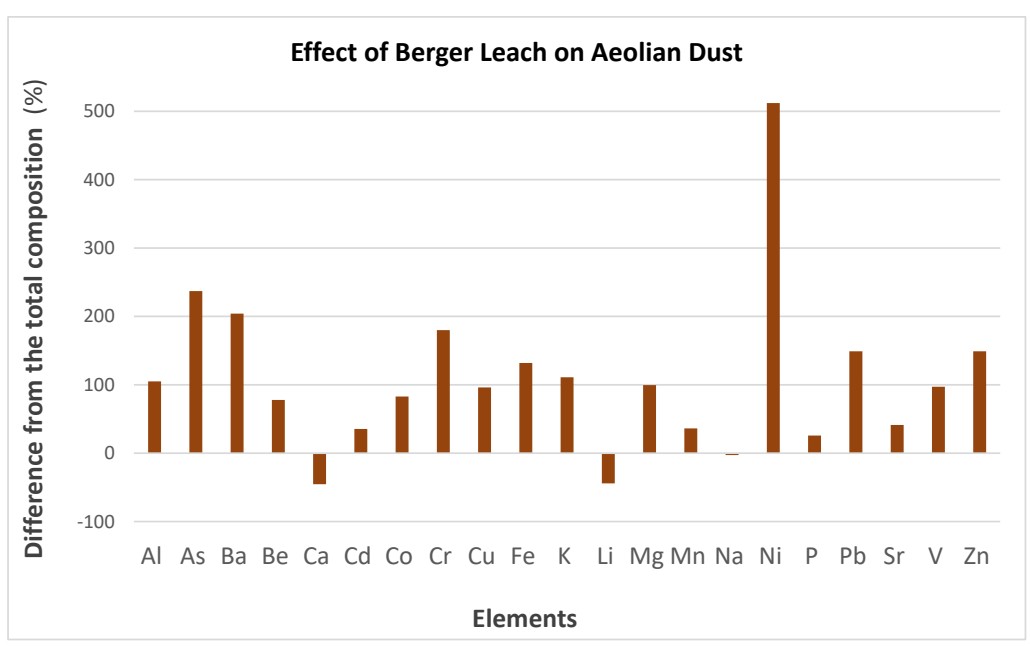





**Figure 4:**

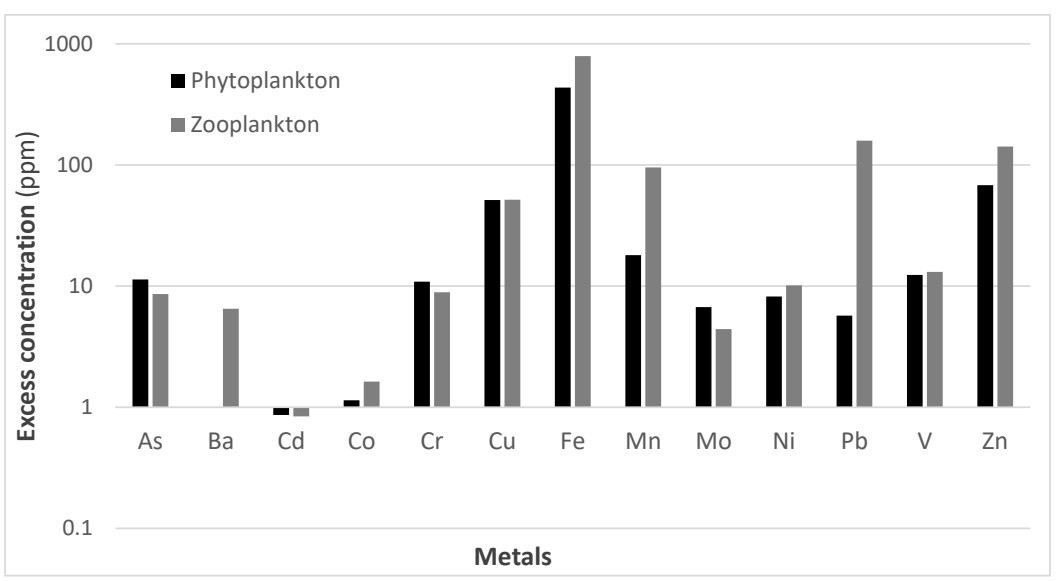



**Figure 5:**

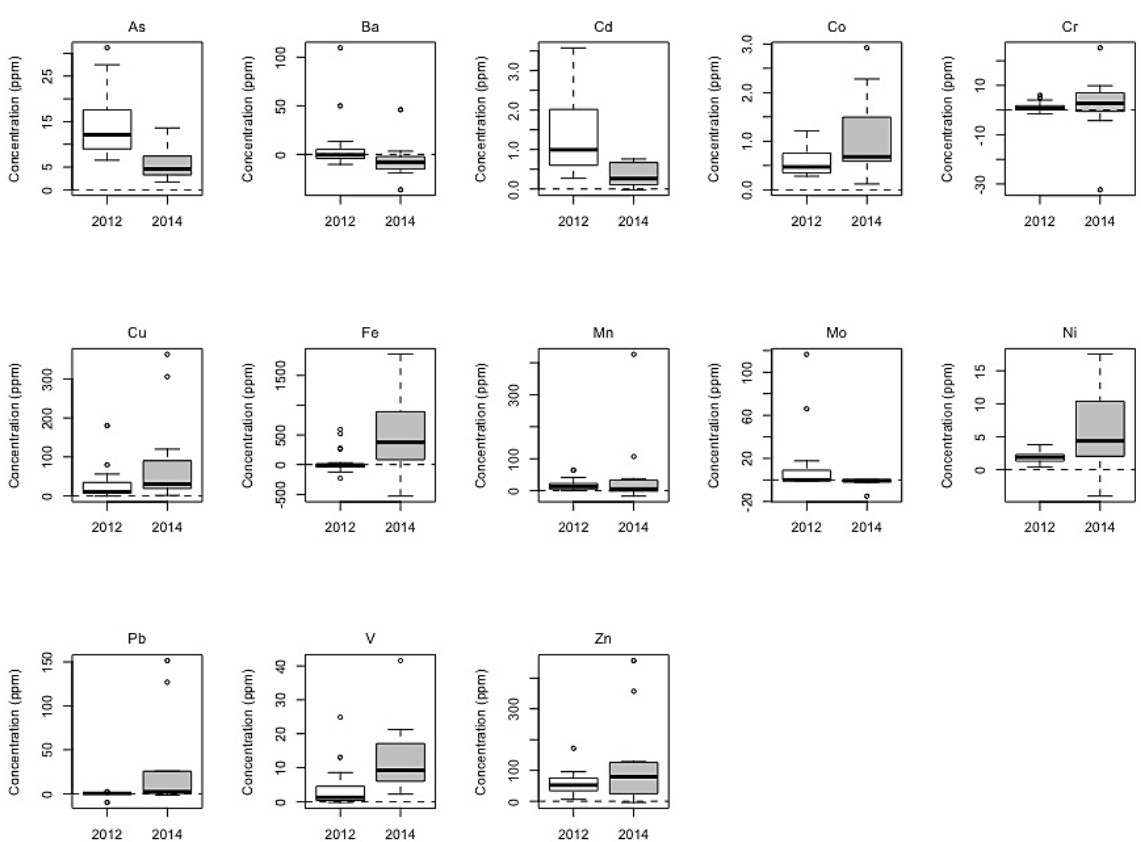





**Figure 6:**

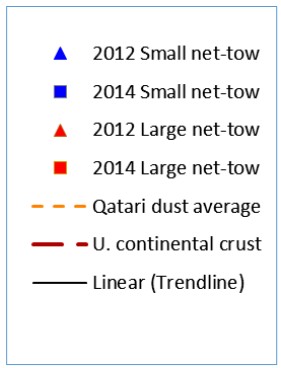

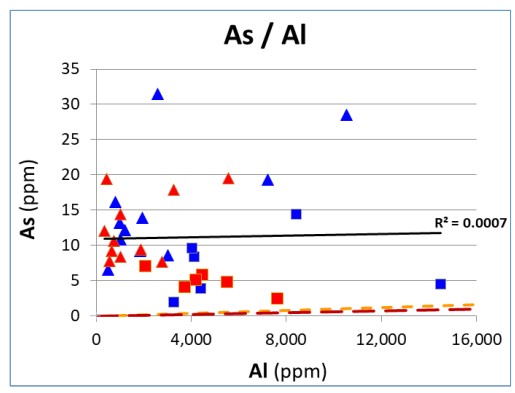

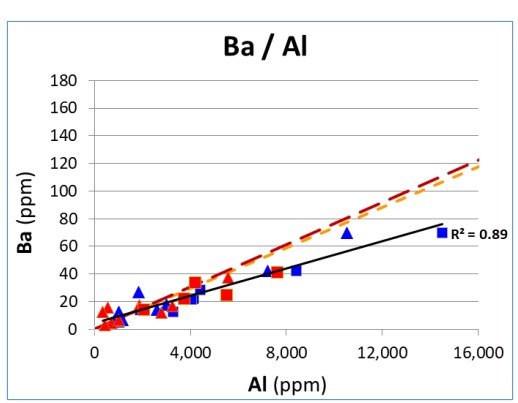

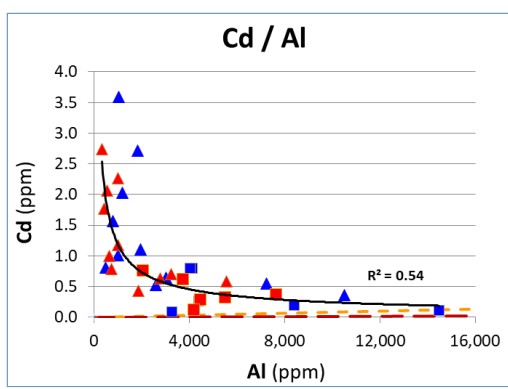

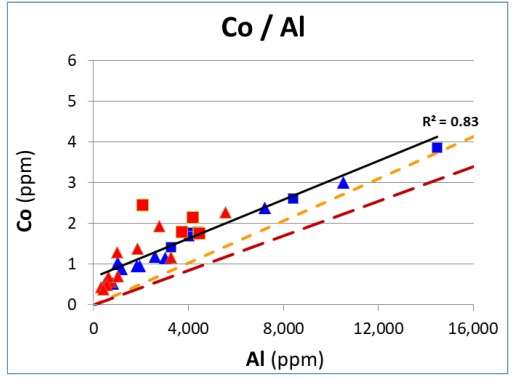

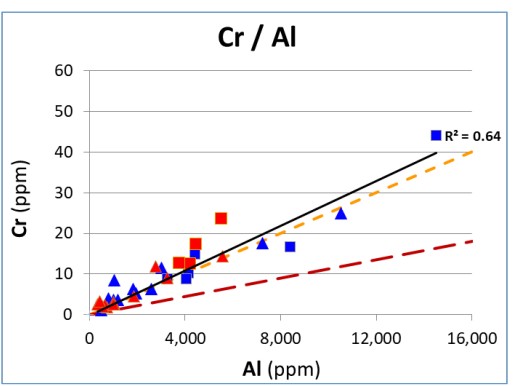




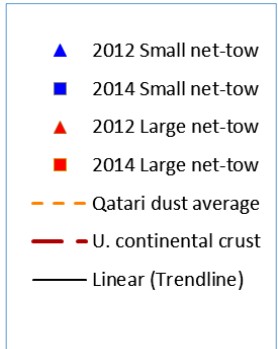

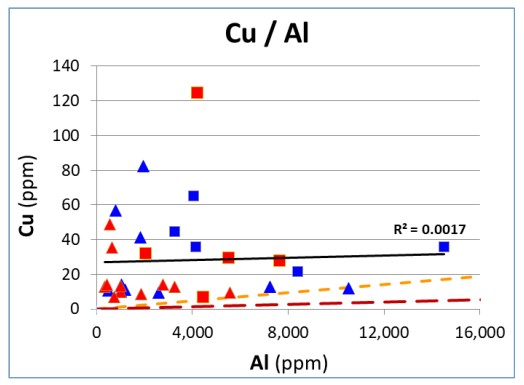

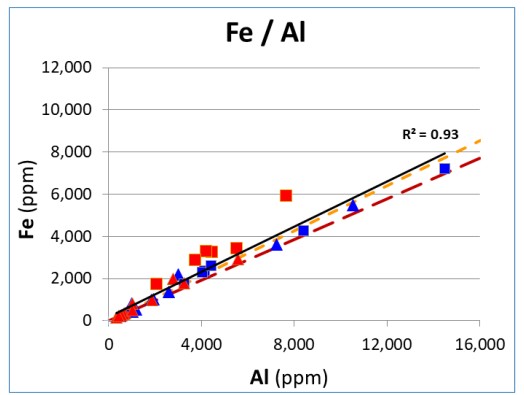

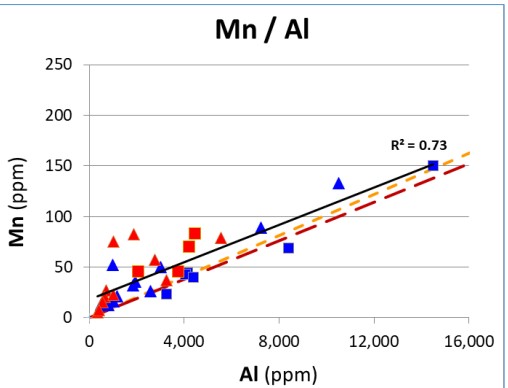

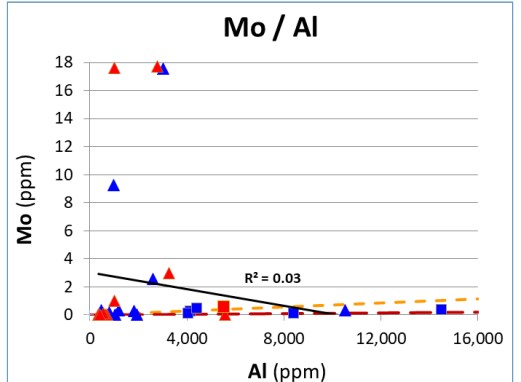

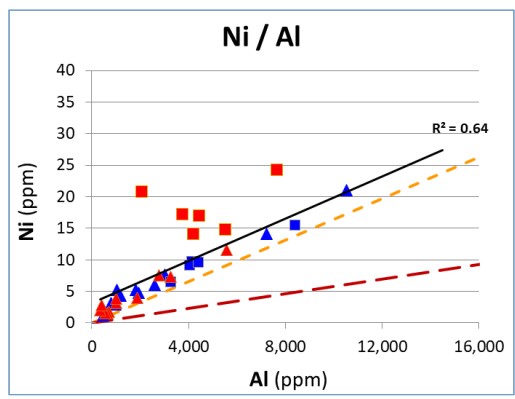





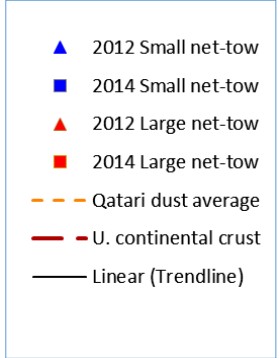

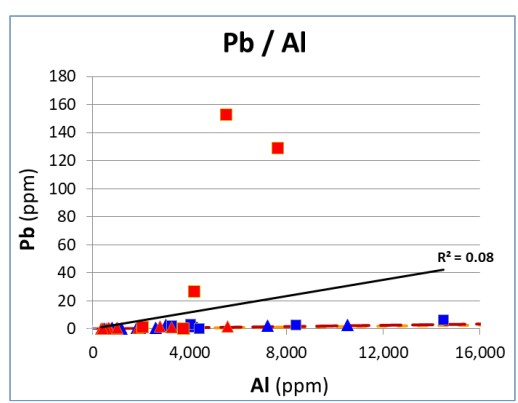

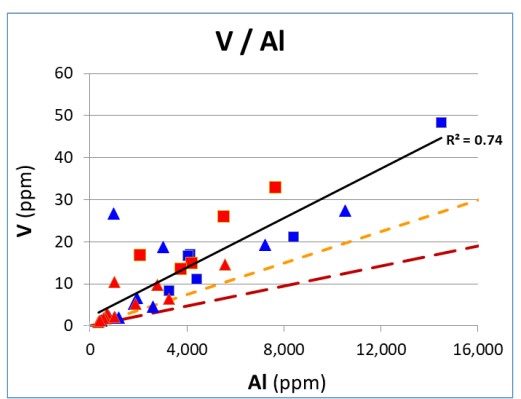

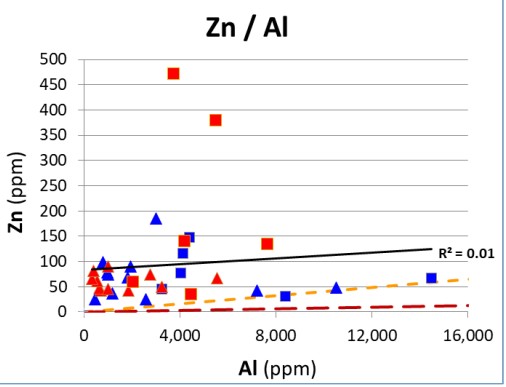

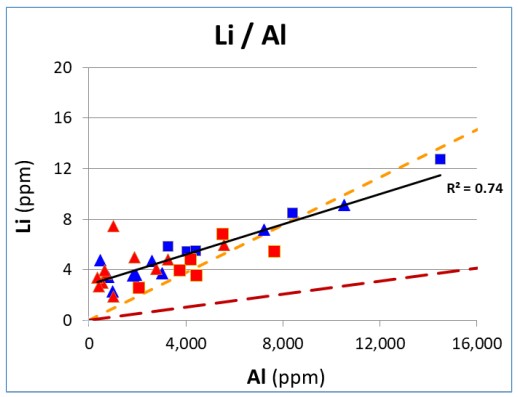

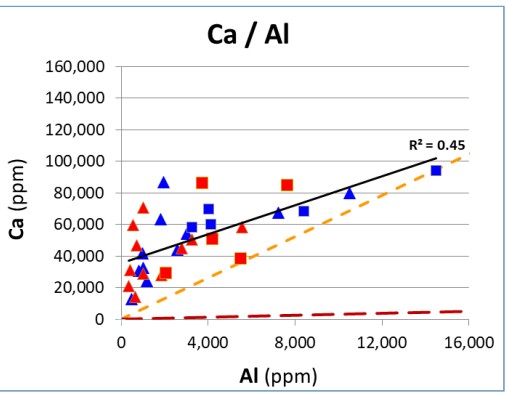



**Figure 7:**

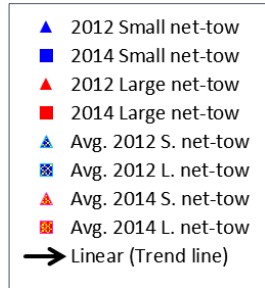

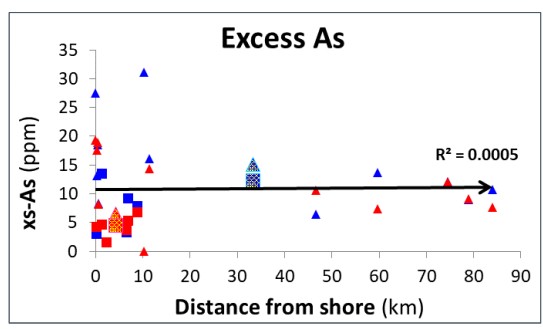

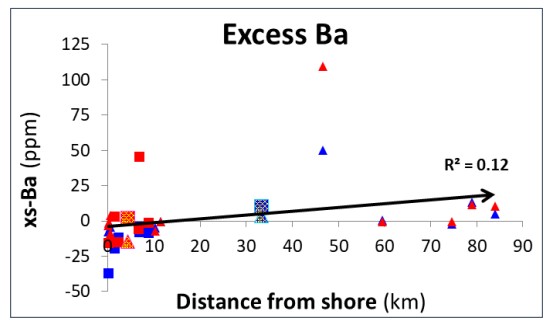

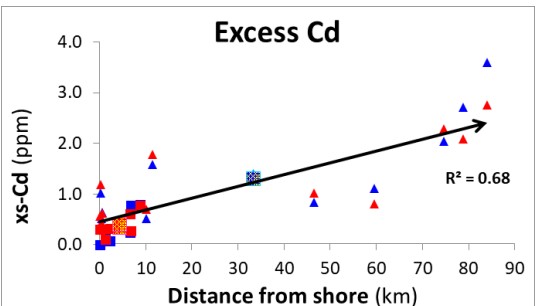

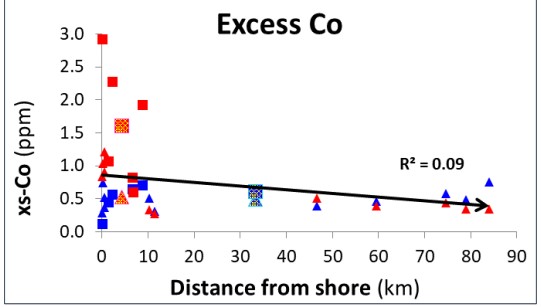

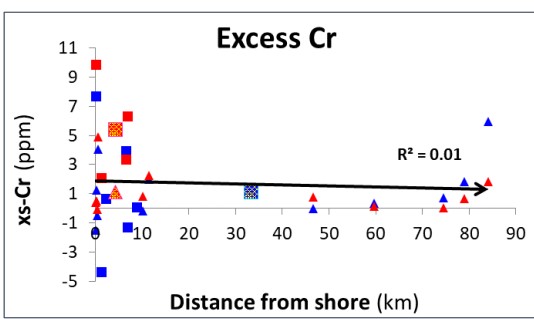




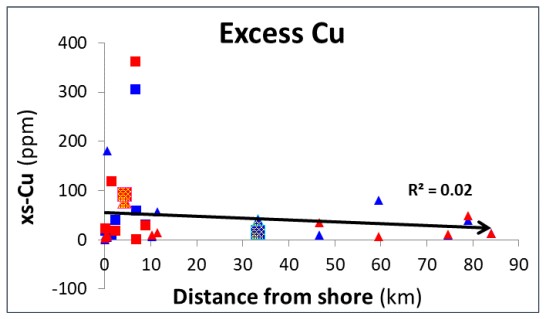
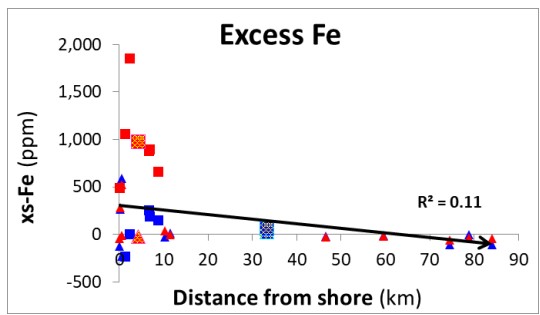

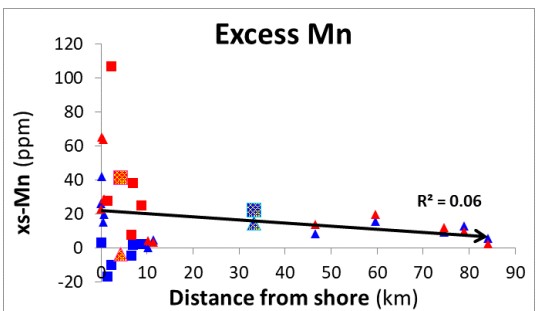
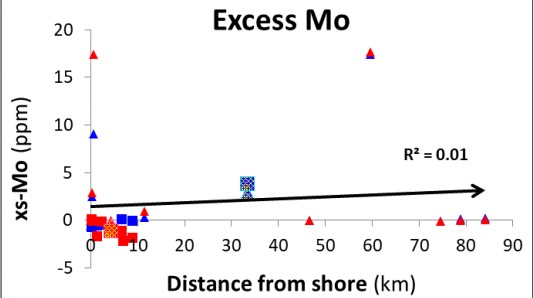

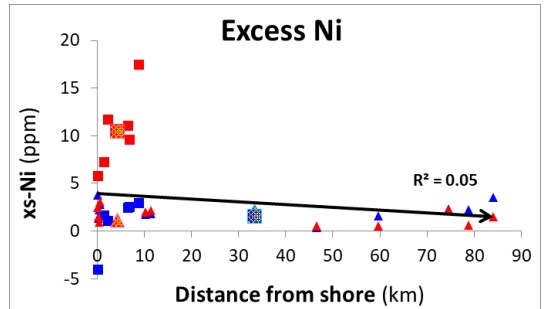
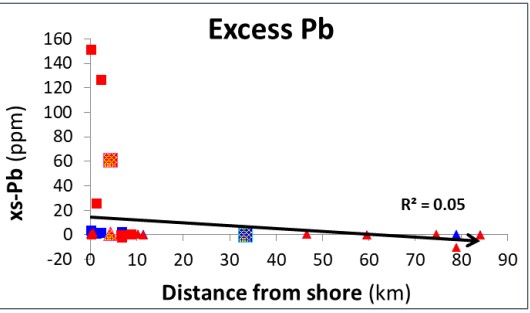

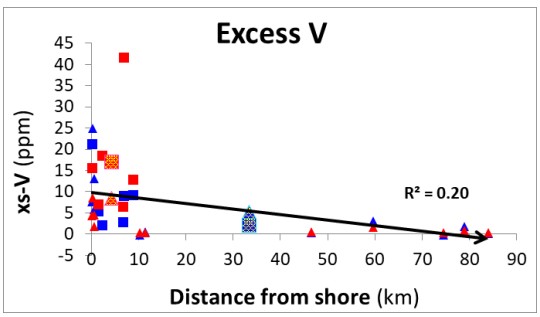
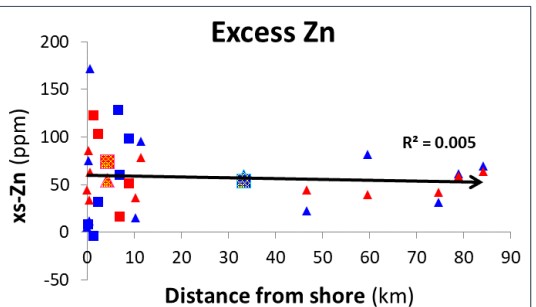



**Tables**

**Table 1:** Average composition of five representative samples of total (strong acid digested, unleached) Qatari dust, comparison with average upper continental crust (UCC; Rudnick and Gao, 2003), and composition of leached (total minus labile) Qatari dust. The total and leached compositions are compared with each other and with UCC as ratios. The unleached and leached compositions are also given as Me/Al ratios.

| Element | Dust (Total)$_{avg}$ | (UCC)$_{R\&G}$ | (Total)$_{avg}$ / (UCC)$_{R\&G}$ | Dust (Leached)$_{avg}$ | (D. Leached)$_{avg}$ / (D. Total)$_{avg}$ | (Leached)$_{avg}$ / (UCC)$_{R\&G}$ | (Me/Al)$_{total}$ | (Me/Al)$_{leach}$ | (Me/Al)$_{UCC}$ |
|---|---|---|---|---|---|---|---|---|---|
| | (ppm) | (ppm) | (ratio) | (ppm) | (ratio) | (ratio) | (ratio) | (ratio) | (ratio) |
| Al | 21,000 | 81,500 | 0.26 | 43,000 | 2.05 | 0.53 | 1 | 1 | 1 |
| As | 2.08 | 4.8 | 0.43 | 7.01 | 3.37 | 1.46 | 9.90E-05 | 1.63E-04 | 5.89E-05 |
| Ba | 154 | 624 | 0.25 | 467 | 3.03 | 0.75 | 7.33E-03 | 1.09E-02 | 7.66E-03 |
| Be | 0.402 | 2.1 | 0.19 | 0.715 | 1.78 | 0.34 | 1.91E-05 | 1.66E-05 | 2.58E-05 |
| Ca | 137,000 | 25,700 | 5.33 | 75,100 | 0.55 | 2.92 | 6.52E+00 | 1.75E+00 | 3.15E-01 |
| Cd | 0.174 | 0.09 | 1.93 | 0.236 | 1.36 | 2.62 | 8.29E-06 | 5.49E-06 | 1.10E-06 |
| Co | 5.42 | 17.3 | 0.31 | 9.91 | 1.83 | 0.57 | 2.58E-04 | 2.30E-04 | 2.12E-04 |
| Cr | 52.6 | 92 | 0.57 | 147 | 2.79 | 1.60 | 2.50E-03 | 3.42E-03 | 1.13E-03 |
| Cu | 24.7 | 28 | 0.88 | 48.4 | 1.96 | 1.73 | 1.18E-03 | 1.13E-03 | 3.44E-04 |
| Fe | 11,200 | 39,200 | 0.29 | 26,100 | 2.33 | 0.67 | 5.33E-01 | 6.07E-01 | 4.81E-01 |
| K | 4,260 | 23,200 | 0.18 | 9,010 | 2.12 | 0.39 | 2.03E-01 | 2.10E-01 | 2.85E-01 |
| Li | 38.3 | 21 | 1.82 | 21.4 | 0.56 | 1.02 | 1.82E-03 | 4.98E-04 | 2.58E-04 |
| Mg | 29,000 | 15,000 | 1.93 | 57,900 | 2.00 | 3.86 | 1.38E+00 | 1.35E+00 | 1.84E-01 |
| Mn | 213 | 775 | 0.27 | 290 | 1.36 | 0.37 | 1.01E-02 | 6.74E-03 | 9.51E-03 |
| Mo | 1.5 | 1.1 | 1.36 | 51.6 | 34.40 | 46.91 | 7.14E-05 | 1.20E-03 | 1.35E-05 |
| Na | 6,550 | 24,300 | 0.27 | 6,370 | 0.97 | 0.26 | 3.12E-01 | 1.48E-01 | 2.98E-01 |
| Ni | 34.5 | 47 | 0.73 | 211 | 6.12 | 4.49 | 1.64E-03 | 4.91E-03 | 5.77E-04 |
| P | 310 | 655 | 0.47 | 389 | 1.25 | 0.59 | 1.48E-02 | 9.05E-03 | 8.04E-03 |
| Pb | 3.64 | 17 | 0.21 | 9.05 | 2.49 | 0.53 | 1.73E-04 | 2.10E-04 | 2.09E-04 |
| Sr | 775 | 320 | 2.42 | 1,100 | 1.42 | 3.44 | 3.69E-02 | 2.56E-02 | 3.93E-03 |
| V | 39.2 | 97 | 0.40 | 77.3 | 1.97 | 0.80 | 1.87E-03 | 1.80E-03 | 1.19E-03 |
| Zn | 85.1 | 67 | 1.27 | 212 | 2.49 | 3.16 | 4.05E-03 | 4.93E-03 | 8.22E-04 |

**Note:** "D." means "Dust"





**Table 2 (next two pages):** Data for the trace element composition of plankton from the *2012 cruise* with average Qatari dust, used for the lithogenic correction. The upper segment shows results for bulk plankton collected with a 50 µm net tow. The lower segment shows data for the larger plankton collected with a 200 µm net tow. The 'CUMULATIVE' row is an average of both of these size fractions. For both size classes and the CUMULATIVE data averages and standard deviations are given. The distance column gives the shortest distance from land to each sampling location (in kilometers). The letters (R), (L), and (E) represent the raw data, the lithogenic correction (in this case with average Qatari dust), and the excess concentrations, respectively. All concentrations are parts per million (ppm), or µg/g. Aluminum only has a raw data column because of the entire Al in each sample was assumed to be of lithogenic origin, meaning that the lithogenic correction would be zero. The number in scientific notation in parenthesis adjacent to the (L) column is a numerical representation of the [Me]/Al ratio found in Qatari dust (Yigiterhan et al, 2018) used to determine the lithogenic correction. Certain elements have excess values, which are negative, indicating that there was an overcorrection for the lithogenic portion of the sample.



**Bulk plankton (50 μm)**

| | | | Al | As | | | Ba | | | Cd | | | Co | | | Cr | | | Cu | | |
|---|---|---|---|---|---|---|---|---|---|---|---|---|---|---|---|---|---|---|---|---|---|
| Sample # | Location | Dist.(km) | R | R | L | E | R | L | E | R | L | E | R | L | E | R | L | E | R | L | E |
| 1 | Doha | 10.19 | 2,592 | 31.4 | 0.26 | 31.2 | 14.1 | 18.97 | -4.92 | 0.52 | 0.02 | 0.50 | 1.18 | 0.67 | 0.51 | 6.31 | 6.49 | -0.18 | 9.3 | 3.04 | 6.2 |
| 2 | Khor Al-Odaid | 0.23 | 994 | 13.2 | 0.10 | 13.1 | 5.7 | 7.27 | -1.59 | 1.01 | 0.01 | 1.00 | 1.00 | 0.26 | 0.75 | 3.72 | 2.49 | 1.23 | 10.4 | 1.17 | 9.2 |
| 3 | Mesaieed | 0.60 | 3,013 | 8.6 | 0.30 | 8.3 | 17.9 | 22.05 | -4.11 | 0.64 | 0.02 | 0.62 | 1.14 | 0.78 | 0.37 | 11.59 | 7.54 | 4.05 | 183.7 | 3.54 | 180.2 |
| 4 | Shraawoo | 59.60 | 1,943 | 13.9 | 0.19 | 13.7 | 14.7 | 14.22 | 0.48 | 1.11 | 0.02 | 1.09 | 0.96 | 0.50 | 0.46 | 5.21 | 4.86 | 0.35 | 82.3 | 2.28 | 80.1 |
| 5 | Al-Edd Al-Gharbi | 46.55 | 480 | 6.6 | 0.05 | 6.5 | 53.4 | 3.52 | 49.84 | 0.81 | 0.00 | 0.81 | 0.52 | 0.12 | 0.39 | 1.20 | 1.20 | 0.00 | 10.8 | 0.56 | 10.2 |
| 6 | Ash Shargi Oilfield | 78.90 | 1,830 | 9.2 | 0.18 | 9.0 | 26.8 | 13.39 | 13.36 | 2.71 | 0.02 | 2.70 | 0.95 | 0.47 | 0.48 | 6.40 | 4.58 | 1.82 | 41.4 | 2.15 | 39.3 |
| 7 | Halul Island | 84.05 | 1,026 | 10.9 | 0.10 | 10.8 | 12.4 | 7.51 | 4.88 | 3.59 | 0.01 | 3.58 | 1.02 | 0.26 | 0.76 | 8.51 | 2.57 | 5.94 | 13.9 | 1.21 | 12.7 |
| 8 | High North | 74.55 | 1,182 | 12.2 | 0.12 | 12.1 | 6.6 | 8.65 | -2.02 | 2.03 | 0.01 | 2.02 | 0.88 | 0.30 | 0.57 | 3.71 | 2.96 | 0.75 | 11.2 | 1.39 | 9.8 |
| 9 | Ras Laffan | 11.40 | 803 | 16.1 | 0.08 | 16.1 | 5.5 | 5.88 | -0.33 | 1.56 | 0.01 | 1.56 | 0.52 | 0.21 | 0.31 | 4.00 | 2.01 | 1.99 | 56.9 | 0.94 | 56.0 |
| 10 | Dukhan | 0.02 | 10,524 | 28.5 | 1.04 | 27.5 | 69.5 | 77.02 | -7.49 | 0.36 | 0.09 | 0.27 | 3.00 | 2.71 | 0.29 | 24.87 | 26.35 | -1.49 | 11.9 | 12.36 | -0.5 |
| 11 | Umm Bab | 0.48 | 7,227 | 19.3 | 0.72 | 18.6 | 42.5 | 52.89 | -10.43 | 0.55 | 0.06 | 0.49 | 2.38 | 1.86 | 0.52 | 17.59 | 18.10 | -0.51 | 12.6 | 8.49 | 4.1 |
| | **Average** | | 2,874 | 15.4 | | 15.2 | 24.5 | | 3.42 | 1.35 | | 1.33 | 1.23 | | 0.49 | 8.47 | | 1.27 | 40.4 | | 37.0 |
| | 2x St.Dev. | | 6,304 | 16.1 | | 15.7 | 43.0 | | 33.28 | 2.06 | | 2.09 | 1.53 | | 0.31 | 14.13 | | 4.28 | 106.7 | | 107.6 |

**Zooplankton (200 μm)**

| | | | R | R | L | E | R | L | E | R | L | E | R | L | E | R | L | E | R | L | E |
|---|---|---|---|---|---|---|---|---|---|---|---|---|---|---|---|---|---|---|---|---|---|
| 12 | Doha | 10.19 | 3,253 | 17.9 | 0.32 | 17.6 | 16.9 | 23.80 | -6.86 | 0.70 | 0.03 | 0.68 | 1.17 | 0.84 | 0.33 | 8.96 | 8.14 | 0.82 | 12.9 | 3.82 | 9.1 |
| 13 | Khor Al-Odaid | 0.23 | 1,012 | 8.4 | 0.10 | 8.3 | 6.1 | 7.40 | -1.34 | 1.17 | 0.01 | 1.17 | 1.30 | 0.26 | 1.04 | 3.07 | 2.53 | 0.54 | 9.8 | 1.19 | 8.6 |
| 14 | Mesaieed | 0.60 | 2,774 | 7.6 | 0.27 | 7.4 | 11.8 | 20.30 | -8.54 | 0.63 | 0.02 | 0.60 | 1.93 | 0.72 | 1.21 | 11.87 | 6.95 | 4.92 | 14.1 | 3.26 | 10.8 |
| 15 | Shraawoo | 59.60 | 722 | 10.7 | 0.07 | 10.6 | 4.8 | 5.28 | -0.52 | 0.78 | 0.01 | 0.78 | 0.58 | 0.19 | 0.39 | 1.93 | 1.81 | 0.12 | 6.8 | 0.85 | 5.9 |
| 16 | Al-Edd Al-Gharbi | 46.55 | 641 | 9.2 | 0.06 | 9.2 | 114.2 | 4.69 | 109.53 | 1.00 | 0.01 | 0.99 | 0.68 | 0.17 | 0.51 | 2.40 | 1.61 | 0.79 | 35.5 | 0.75 | 34.8 |
| 17 | Ash Shargi Oilfield | 78.90 | 560 | 7.8 | 0.06 | 7.7 | 15.6 | 4.09 | 11.55 | 2.07 | 0.00 | 2.06 | 0.48 | 0.14 | 0.34 | 2.09 | 1.40 | 0.69 | 48.9 | 0.66 | 48.2 |
| 18 | Halul Island | 84.05 | 335 | 12.1 | 0.03 | 12.1 | 12.8 | 2.45 | 10.33 | 2.74 | 0.00 | 2.74 | 0.43 | 0.09 | 0.35 | 2.65 | 0.84 | 1.81 | 12.7 | 0.39 | 12.3 |
| 19 | High North | 74.55 | 1,019 | 14.5 | 0.10 | 14.4 | 6.7 | 7.46 | -0.78 | 2.27 | 0.01 | 2.26 | 0.70 | 0.26 | 0.44 | 2.61 | 2.55 | 0.06 | 13.1 | 1.20 | 11.9 |
| 20 | Ras Laffan | 11.40 | 418 | 19.4 | 0.04 | 19.4 | 2.8 | 3.06 | -0.28 | 1.77 | 0.00 | 1.76 | 0.39 | 0.11 | 0.28 | 3.29 | 1.05 | 2.25 | 14.1 | 0.49 | 13.6 |
| 21 | Dukhan | 0.02 | 5,560 | 19.5 | 0.55 | 19.0 | 37.6 | 40.69 | -3.12 | 0.59 | 0.05 | 0.54 | 2.28 | 1.43 | 0.84 | 14.35 | 13.92 | 0.43 | 9.5 | 6.53 | 3.0 |
| 22 | Umm Bab | 0.48 | 1,865 | 9.5 | 0.18 | 9.3 | 17.5 | 13.65 | 3.83 | 0.43 | 0.02 | 0.42 | 1.39 | 0.48 | 0.91 | 4.62 | 4.67 | -0.05 | 8.7 | 2.19 | 6.6 |
| | **Average** | | 1,651 | 12.4 | | 12.2 | 22.4 | | 10.34 | 1.29 | | 1.27 | 1.03 | | 0.60 | 5.26 | | 1.13 | 16.9 | | 15.0 |
| | 2x St.Dev. | | 3,243 | 9.3 | | 9.2 | 63.8 | | 66.96 | 1.58 | | 1.60 | 1.28 | | 0.66 | 8.77 | | 2.90 | 26.1 | | 27.6 |
| | **Cumulative** | | 2,262 | 13.9 | | 13.7 | 23.4 | | 6.88 | 1.32 | | 1.30 | 1.13 | | 0.55 | 6.86 | | 1.20 | 28.7 | | 26.0 |
| | 2x St.Dev. | | 5,050 | 13.2 | | 12.9 | 53.1 | | 52.08 | 1.80 | | 1.82 | 1.39 | | 0.52 | 11.94 | | 3.58 | 79.5 | | 79.9 |



Bulkplankton (50 µm)

| Sample # | Location | Dist.(km) | Fe R | Fe L | Fe E | Mn R | Mn L | Mn E | Mo R | Mo L | Mo E | Ni R | Ni L | Ni E | Pb R | Pb L | Pb E | V R | V L | V E | Zn R | Zn L | Zn E |
|---|---|---|---|---|---|---|---|---|---|---|---|---|---|---|---|---|---|---|---|---|---|---|---|
| 1 | Doha | 10.19 | 1,359 | 1,384 | -24.4 | 26.3 | 26.3 | -0.02 | 116.86 | 0.18 | 116.67 | 5.99 | 4.26 | 1.73 | 0.70 | 0.45 | 0.25 | 4.59 | 4.84 | -0.25 | 25.5 | 10.5 | 15.0 |
| 2 | Khor Al-Odaid | 0.23 | 793 | 530 | 262.7 | 52.2 | 10.1 | 42.1 | 2.59 | 0.07 | 2.52 | 2.94 | 1.63 | 1.31 | 0.34 | 0.17 | 0.17 | 26.74 | 1.86 | 24.89 | 78.9 | 4.0 | 74.9 |
| 3 | Mesaieed | 0.60 | 2,200 | 1,608 | 592.1 | 50.1 | 30.5 | 19.5 | 9.26 | 0.21 | 9.04 | 7.71 | 4.95 | 2.76 | 3.01 | 0.52 | 2.49 | 18.70 | 5.63 | 13.08 | 184.5 | 12.2 | 172.3 |
| 4 | Shraawoo | 59.60 | 1,023 | 1,037 | -14.0 | 35.5 | 19.7 | 15.8 | 17.58 | 0.14 | 17.44 | 4.77 | 3.19 | 1.58 | 0.50 | 0.34 | 0.16 | 6.62 | 3.63 | 2.99 | 89.8 | 7.9 | 82.0 |
| 5 | Al-Edd Al-Gharbi | 46.55 | 239 | 256 | -17.3 | 13.2 | 4.9 | 8.3 | 0.00 | 0.03 | -0.03 | 1.17 | 0.79 | 0.38 | 0.33 | 0.08 | 0.25 | 1.27 | 0.90 | 0.37 | 24.5 | 1.9 | 22.6 |
| 6 | Ash Shargi Oilfield | 78.90 | 977 | 977 | 0.8 | 31.7 | 18.5 | 13.1 | 0.31 | 0.13 | 0.18 | 5.27 | 3.01 | 2.26 | 0.80 | 0.32 | 0.48 | 5.27 | 3.42 | 1.85 | 68.7 | 7.4 | 61.3 |
| 7 | Halul Island | 84.05 | 437 | 548 | -110.6 | 16.0 | 10.4 | 5.6 | 0.27 | 0.07 | 0.19 | 5.20 | 1.69 | 3.51 | 0.19 | 0.18 | 0.01 | 2.05 | 1.92 | 0.13 | 73.7 | 4.2 | 69.6 |
| 8 | High North | 74.55 | 526 | 631 | -104.9 | 21.5 | 12.0 | 9.5 | 0.00 | 0.08 | -0.08 | 4.26 | 1.94 | 2.32 | 0.31 | 0.20 | 0.11 | 1.97 | 2.21 | -0.24 | 36.6 | 4.8 | 31.8 |
| 9 | Ras Laffan | 11.40 | 445 | 429 | 16.6 | 12.8 | 8.1 | 4.6 | 0.34 | 0.06 | 0.28 | 3.12 | 1.32 | 1.80 | 0.80 | 0.14 | 0.66 | 2.08 | 1.50 | 0.58 | 98.8 | 3.3 | 95.5 |
| 10 | Dukhan | 0.02 | 5,493 | 5,617 | -123.2 | 132.7 | 106.7 | 26.1 | 0.18 | 0.75 | -0.57 | 21.06 | 17.30 | 3.77 | 3.03 | 1.82 | 1.20 | 27.31 | 19.66 | 7.66 | 48.5 | 42.7 | 5.8 |
| 11 | Umm Bab | 0.48 | 3,628 | 3,857 | -229.3 | 88.6 | 73.3 | 15.3 | 0.32 | 0.51 | -0.20 | 14.11 | 11.88 | 2.23 | 2.12 | 1.25 | 0.87 | 19.28 | 13.50 | 5.78 | 41.7 | 29.3 | 12.4 |
| | **Average** | | **1,556** | | **22.6** | **43.7** | | **14.5** | **13.43** | | **13.22** | **6.87** | | **2.15** | **1.10** | | **0.60** | **10.54** | | **5.17** | **70.1** | | **58.5** |
| | 2x St.Dev. | | 3,270 | | 450.0 | 74.1 | | 23.4 | 69.50 | | 69.52 | 11.56 | | 1.93 | 2.17 | | 1.45 | 20.67 | | 15.52 | 91.3 | | 98.1 |

Zooplankton (200 µm)

| Sample # | Location | Dist.(km) | Fe R | Fe L | Fe E | Mn R | Mn L | Mn E | Mo R | Mo L | Mo E | Ni R | Ni L | Ni E | Pb R | Pb L | Pb E | V R | V L | V E | Zn R | Zn L | Zn E |
|---|---|---|---|---|---|---|---|---|---|---|---|---|---|---|---|---|---|---|---|---|---|---|---|
| 12 | Doha | 10.19 | 1,770 | 1,736 | 34.2 | 37.2 | 33.0 | 4.2 | 66.15 | 0.23 | 65.92 | 7.30 | 5.35 | 1.96 | 1.15 | 0.56 | 0.58 | 6.50 | 6.08 | 0.42 | 49.2 | 13.2 | 36.0 |
| 13 | Khor Al-Odaid | 0.23 | 817 | 540 | 277.5 | 75.3 | 10.3 | 65.0 | 2.98 | 0.07 | 2.90 | 3.15 | 1.66 | 1.49 | 0.18 | 0.18 | 0.01 | 10.47 | 1.89 | 8.58 | 90.1 | 4.1 | 86.0 |
| 14 | Mesaieed | 0.60 | 2,001 | 1,480 | 521.0 | 57.1 | 28.1 | 29.0 | 17.63 | 0.20 | 17.44 | 7.57 | 4.56 | 3.01 | 2.03 | 0.48 | 1.55 | 9.69 | 5.18 | 4.51 | 74.6 | 11.2 | 63.3 |
| 15 | Shraawoo | 59.60 | 370 | 385 | -15.7 | 27.0 | 7.3 | 19.7 | 17.74 | 0.05 | 17.69 | 1.73 | 1.19 | 0.54 | 0.16 | 0.13 | 0.04 | 2.99 | 1.35 | 1.64 | 42.3 | 2.9 | 39.3 |
| 16 | Al-Edd Al-Gharbi | 46.55 | 315 | 342 | -26.9 | 20.3 | 6.5 | 13.8 | 0.00 | 0.05 | -0.05 | 1.58 | 1.05 | 0.52 | 0.56 | 0.11 | 0.45 | 1.80 | 1.20 | 0.60 | 46.7 | 2.6 | 44.1 |
| 17 | Ash Shargi Oilfield | 78.90 | 262 | 299 | -36.8 | 15.3 | 5.7 | 9.7 | 0.00 | 0.04 | -0.04 | 1.55 | 0.92 | 0.63 | 0.00 | 0.10 | -10.00 | 1.88 | 1.04 | 0.84 | 61.2 | 2.3 | 59.5 |
| 18 | Halul Island | 84.05 | 136 | 179 | -42.4 | 5.7 | 3.4 | 2.3 | 0.10 | 0.02 | 0.08 | 2.03 | 0.55 | 1.48 | 0.14 | 0.06 | 0.08 | 0.91 | 0.62 | 0.28 | 64.9 | 1.4 | 63.6 |
| 19 | High North | 74.55 | 484 | 544 | -59.9 | 22.5 | 10.3 | 12.1 | 0.00 | 0.07 | -0.07 | 3.90 | 1.67 | 2.23 | 0.57 | 0.18 | 0.39 | 2.17 | 1.90 | 0.27 | 45.9 | 4.1 | 41.8 |
| 20 | Ras Laffan | 11.40 | 222 | 223 | -1.3 | 7.7 | 4.2 | 3.5 | 0.98 | 0.03 | 0.95 | 2.80 | 0.69 | 2.11 | 0.00 | 0.07 | -0.07 | 1.22 | 0.78 | 0.44 | 80.6 | 1.7 | 78.9 |
| 21 | Dukhan | 0.02 | 2,923 | 2,967 | -44.0 | 79.2 | 56.4 | 22.9 | 0.00 | 0.40 | -0.04 | 11.60 | 9.14 | 2.46 | 1.58 | 0.96 | 0.62 | 14.67 | 10.38 | 4.29 | 67.3 | 22.5 | 44.8 |
| 22 | Umm Bab | 0.48 | 988 | 996 | -7.3 | 82.5 | 18.9 | 63.6 | 0.00 | 0.13 | -0.13 | 3.98 | 3.07 | 0.92 | 0.89 | 0.32 | 0.57 | 5.30 | 3.48 | 1.82 | 41.8 | 7.6 | 34.2 |
| | **Average** | | **935** | | **54.4** | **39.1** | | **22.3** | **9.60** | | **9.48** | **4.29** | | **1.58** | **0.66** | | **0.37** | **5.24** | | **2.15** | **60.5** | | **53.8** |
| | 2x St.Dev. | | 1,823 | | 361.6 | 58.6 | | 44.7 | 39.99 | | 39.92 | 6.43 | | 1.70 | 1.36 | | 0.96 | 9.17 | | 5.25 | 33.2 | | 35.3 |
| | **Cumulative** | | **1,246** | | **38.5** | **41.4** | | **18.4** | **11.51** | | **11.35** | **5.58** | | **1.86** | **0.88** | | **0.49** | **7.89** | | **3.66** | **65.3** | | **56.1** |
| | 2x St.Dev. | | 2,661 | | 399.6 | 65.3 | | 35.8 | 55.47 | | 55.44 | 9.50 | | 1.87 | 1.82 | | 1.22 | 16.52 | | 11.72 | 67.8 | | 72.1 |




**Table 3 (next two pages):** Data for the trace element composition of plankton from the *2014 cruise*. The upper segment, data for the bulk plankton collected with a 50 µm net tow, is separated from the lower segment with data for the larger fraction of plankton, collected with a 200 µm net tow.

The 'CUMULATIVE' row is an average of both of these measurements. For both size classes and the CUMULATIVE data averages and standard deviations are given. The distance column is a calculation of the shortest distance from land to each sampling location (in kilometers), The letters (R), (L), and (E) represent the values for raw data, the lithogenic correction (using average Qatar dust), and the excess concentrations, respectively. All concentrations are parts per million (ppm),

or µg/g. Aluminum only has a raw data column because all of the aluminum in each sample was assumed to be of lithogenic origin, meaning that the lithogenic correction would be zero. The number in scientific notation in parenthesis adjacent to the (L) column is a numerical representation of the [Me]/Al ratio found in average Qatari dust (Table 1 and Yigiterhan et al., 2018) used to determine the lithogenic correction. Certain elements have excess values, which are negative,

indicating that there was an overcorrection for the lithogenic portion of the sample.




**Bulkplankton (50 µm)**

| Sample | Location | Dist.(km) | Al R | As R | As L | As E | Ba R | Ba L | Ba E | Cd R | Cd L | Cd E | Co R | Co L | Co E | Cr R | Cr L | Cr E | Cu R | Cu L | Cu E |
|---|---|---|---|---|---|---|---|---|---|---|---|---|---|---|---|---|---|---|---|---|---|
| 22 | Dukhan | 8.77 | 4,129 | 8.40 | 0.41 | 7.99 | 21.9 | 30.2 | -8.29 | 0.80 | 0.03 | 0.76 | 1.77 | 1.06 | 0.71 | 10.3 | 10.2 | 0.08 | 36.0 | 4.85 | 31.1 |
| 23 | Dukhan | 6.78 | 4,035 | 9.62 | 0.40 | 9.22 | 21.7 | 29.5 | -7.88 | 0.79 | 0.03 | 0.76 | 1.68 | 1.04 | 0.63 | 8.8 | 10.1 | -1.27 | 65.2 | 4.74 | 60.5 |
| 24 | Dukhan | 1.31 | 8,407 | 14.45 | 0.83 | 13.61 | 42.2 | 61.5 | -19.32 | 0.19 | 0.07 | 0.12 | 2.62 | 2.17 | 0.45 | 16.7 | 21.1 | -4.32 | 21.7 | 9.87 | 11.8 |
| 25 | Doha | 6.54 | 4,401 | 3.87 | 0.44 | 3.43 | 28.6 | 32.2 | -3.61 | 0.26 | 0.04 | 0.22 | 1.78 | 1.13 | 0.65 | 15.0 | 11.0 | 3.94 | 311.2 | 5.17 | 306.0 |
| 26 | Doha | 2.20 | 3,266 | 2.01 | 0.32 | 1.68 | 12.5 | 23.9 | -11.37 | 0.09 | 0.03 | 0.06 | 1.41 | 0.84 | 0.57 | 8.9 | 8.2 | 0.69 | 44.8 | 3.84 | 41.0 |
| 27 | Doha | 0.06 | 14,493 | 4.54 | 1.44 | 3.11 | 69.5 | 106.1 | -36.58 | 0.10 | 0.12 | -0.02 | 3.86 | 3.74 | 0.12 | 44.0 | 36.3 | 7.70 | 35.9 | 17.02 | 18.9 |
| **Average** | | | **6,455** | **7.15** | | **6.51** | **32.7** | | **-14.51** | **0.37** | | **0.32** | **2.19** | | **0.52** | **17.3** | | **1.14** | **85.8** | | **78.2** |
| 2x St.Dev. | | | 8,675 | 9.16 | | 9.14 | 41.0 | | 24.04 | 0.67 | | 0.70 | 1.83 | | 0.43 | 27.0 | | 8.38 | 222.7 | | 225.8 |

**Zooplankton (200 µm)**

| Sample | Location | Dist.(km) | Al R | As R | As L | As E | Ba R | Ba L | Ba E | Cd R | Cd L | Cd E | Co R | Co L | Co E | Cr R | Cr L | Cr E | Cu R | Cu L | Cu E |
|---|---|---|---|---|---|---|---|---|---|---|---|---|---|---|---|---|---|---|---|---|---|
| 28 | Dukhan | 8.77 | 2,054 | 7.04 | 0.20 | 6.83 | 14.1 | 15.0 | -0.90 | 0.77 | 0.02 | 0.75 | 2.45 | 0.53 | 1.92 | -27.2 | 5.1 | -32.36 | 31.9 | 2.41 | 29.5 |
| 29 | Dukhan | 6.78 | 4,448 | 5.83 | 0.44 | 5.39 | 78.4 | 32.6 | 45.83 | 0.29 | 0.04 | 0.25 | 1.75 | 1.15 | 0.60 | 17.5 | 11.1 | 6.34 | 6.9 | 5.22 | 1.6 |
| 30 | Dukhan | 1.31 | 4,190 | 5.17 | 0.41 | 4.76 | 33.9 | 30.7 | 3.19 | 0.12 | 0.03 | 0.09 | 2.15 | 1.08 | 1.07 | 12.6 | 10.5 | 2.08 | 124.8 | 4.92 | 119.9 |
| 31 | Doha | 6.54 | 3,729 | 4.10 | 0.37 | 3.73 | 22.0 | 27.3 | -5.28 | 0.62 | 0.03 | 0.59 | 1.79 | 0.96 | 0.82 | 12.7 | 9.3 | 3.39 | 367.6 | 4.38 | 363.3 |
| 32 | Doha | 2.20 | 7,634 | 2.45 | 0.76 | 1.69 | 40.9 | 55.9 | -14.97 | 0.37 | 0.06 | 0.30 | 4.25 | 1.97 | 2.28 | 44.4 | 19.1 | 25.23 | 28.0 | 8.97 | 19.1 |
| 33 | Doha | 0.06 | 5,506 | 4.86 | 0.55 | 4.32 | 24.8 | 40.3 | -15.49 | 0.33 | 0.05 | 0.28 | 4.34 | 1.42 | 2.92 | 23.7 | 13.8 | 9.88 | 29.5 | 6.47 | 23.0 |
| **Average** | | | **4,594** | **4.91** | | **4.45** | **35.7** | | **2.06** | **0.41** | | **0.38** | **2.79** | | **1.60** | **18.5** | | **2.43** | **98.1** | | **92.7** |
| 2x St.Dev. | | | 3,737 | 3.12 | | 3.44 | 45.8 | | 45.42 | 0.47 | | 0.49 | 2.39 | | 1.83 | 29.8 | | 37.98 | 276.7 | | 277.9 |

| | | | Al R | As R | As L | As E | Ba R | Ba L | Ba E | Cd R | Cd L | Cd E | Co R | Co L | Co E | Cr R | Cr L | Cr E | Cu R | Cu L | Cu E |
|---|---|---|---|---|---|---|---|---|---|---|---|---|---|---|---|---|---|---|---|---|---|
| **Cumulative** | | | **5,524** | **6.03** | | **5.48** | **34.2** | | **-6.22** | **0.39** | | **0.35** | **2.49** | | **1.06** | **17.9** | | **1.78** | **92.0** | | **85.5** |
| 2x St.Dev. | | | 6,659 | 6.93 | | 6.92 | 41.6 | | 42.18 | 0.55 | | 0.58 | 2.12 | | 1.70 | 27.1 | | 26.26 | 239.8 | | 241.9 |




Bulkplankton (50 µm)

| Sample | Location | Dist.(km) | Fe R | Fe L | Fe E | Mn R | Mn L | Mn E | Mo R | Mo L | Mo E | Ni R | Ni L | Ni E | Pb R | Pb L | Pb E | V R | V L | V E | Zn R | Zn L | Zn E |
|---|---|---|---|---|---|---|---|---|---|---|---|---|---|---|---|---|---|---|---|---|---|---|---|
| 22 | Dukhan | 8.77 | 2,358 | 2,204 | 155 | 44.1 | 41.85 | 2.3 | 0.29 | 0.29 | 0.002 | 9.7 | 6.79 | 2.9 | 1.3 | 0.72 | 0.6 | 17.0 | 7.71 | 9.3 | 116 | 16.74 | 99 |
| 23 | Dukhan | 6.78 | 2,305 | 2,111 | 194 | 43.1 | 40.90 | 2.2 | 0.14 | 0.29 | -15.00 | 9.2 | 6.63 | 2.6 | 3.4 | 0.70 | 2.7 | 16.7 | 7.54 | 9.2 | 77 | 16.35 | 60 |
| 24 | Dukhan | 1.31 | 4,258 | 4,486 | -228 | 68.5 | 85.21 | -16.8 | 0.10 | 0.59 | -0.49 | 15.5 | 13.82 | 1.7 | 2.7 | 1.46 | 1.3 | 21.2 | 15.70 | 5.5 | 30 | 34.07 | -4 |
| 25 | Doha | 6.54 | 2,603 | 2,349 | 255 | 40.1 | 44.60 | -4.5 | 0.48 | 0.31 | 0.17 | 9.7 | 7.23 | 2.4 | _ | 0.76 | _ | 11.1 | 8.22 | 2.9 | 147 | 17.84 | 129 |
| 26 | Doha | 2.20 | 1,752 | 1,743 | 9 | 23.4 | 33.11 | -9.7 | -0.19 | 0.23 | -0.43 | 6.5 | 5.37 | 1.1 | 2.2 | 0.57 | 1.7 | 8.3 | 6.10 | 2.2 | 45 | 13.24 | 32 |
| 27 | Doha | 0.06 | 7,205 | 7,735 | -530 | 150.2 | 146.90 | 3.3 | 0.40 | 1.03 | -0.63 | 19.8 | 23.82 | -4.0 | 6.2 | 2.51 | 3.7 | 48.4 | 27.07 | 21.3 | 67 | 58.75 | 9 |
| Average | | | 3,414 | | -24 | 61.6 | | -3.9 | 0.20 | | -0.25 | 11.7 | | 1.1 | 3.2 | | 2.0 | 20.4 | | 8.4 | 80 | | 54 |
| 2x St.Dev. | | | 4,084 | | 604 | 91.5 | | 16.2 | 0.49 | | 0.62 | 9.8 | | 5.2 | 3.7 | | 2.4 | 28.9 | | 14.0 | 88 | | 104 |

Zooplankton (200 µm)

| Sample | Location | Dist.(km) | Fe R | Fe L | Fe E | Mn R | Mn L | Mn E | Mo R | Mo L | Mo E | Ni R | Ni L | Ni E | Pb R | Pb L | Pb E | V R | V L | V E | Zn R | Zn L | Zn E |
|---|---|---|---|---|---|---|---|---|---|---|---|---|---|---|---|---|---|---|---|---|---|---|---|
| 28 | Dukhan | 8.77 | 1,757 | 1,096 | 661 | 45.8 | 20.82 | 25.0 | -1.67 | 0.15 | -1.82 | 20.9 | 3.38 | 17.5 | 1.1 | 0.36 | 0.6 | 16.8 | 3.84 | 13.0 | 60 | 8.33 | 52 |
| 29 | Dukhan | 6.78 | 3,275 | 2,374 | 902 | 83.4 | 45.08 | 38.3 | -1.78 | 0.32 | -2.10 | 16.9 | 7.31 | 9.6 | -0.7 | 0.77 | -1.8 | 49.9 | 8.31 | 41.6 | 35 | 18.03 | 17 |
| 30 | Dukhan | 1.31 | 3,299 | 2,236 | 1,063 | 70.3 | 42.46 | 27.8 | -1.33 | 0.30 | -1.63 | 14.1 | 6.89 | 7.2 | 26.7 | 0.73 | 25.7 | 14.9 | 7.82 | 7.1 | 140 | 16.98 | 123 |
| 31 | Doha | 6.54 | 2,874 | 1,990 | 884 | 45.6 | 37.80 | 7.8 | -0.80 | 0.27 | -1.06 | 17.2 | 6.13 | 11.1 | _ | 0.65 | _ | 13.5 | 6.96 | 6.5 | 472 | 15.12 | 457 |
| 32 | Doha | 2.20 | 5,932 | 4,074 | 1,858 | 184.4 | 77.38 | 107.0 | -0.11 | 0.54 | -0.07 | 24.3 | 12.55 | 11.7 | 128.8 | 1.32 | 126.9 | 32.9 | 14.26 | 18.6 | 134 | 30.94 | 103 |
| 33 | Doha | 0.06 | 3,434 | 2,939 | 495 | 482.5 | 55.81 | 426.6 | 0.57 | 0.39 | 0.18 | 14.9 | 9.05 | 5.8 | 152.9 | 0.95 | 151.6 | 26.0 | 10.28 | 15.7 | 380 | 22.32 | 357 |
| Average | | | 3,429 | | 977 | 152.0 | | 105.4 | 0.10 | | -1.18 | 18.0 | | 10.5 | 61.9 | | 60.6 | 25.7 | | 17.1 | 204 | | 185 |
| 2x St.Dev. | | | 2,743 | | 951 | 339.7 | | 322.1 | 0.46 | | 0.15 | 7.7 | | 8.2 | 146.7 | | 146.2 | 28.0 | | 25.8 | 359 | | 358 |
| | | | | | | | | | | | | | | | | | | | | | | | |
| Cumulative | | | 3,421 | | 452 | 106.8 | | 50.8 | 0.15 | | -0.72 | 14.9 | | 5.8 | 32.5 | | 31.3 | 23.1 | | 12.7 | 142 | | 120 |
| 2x St.Dev. | | | 3,317 | | 1,292 | 255.3 | | 245.6 | 0.47 | | 1.56 | 10.7 | | 11.8 | 115.8 | | 115.4 | 27.7 | | 21.8 | 280 | | 286 |



**Table 4:** Are dust corrected plankton concentrations statistically different from zero for 2012 data?

| Element | > Zero | p_value | Shapiro | not_normal | test_used | t_test | wilcoxon |
|---------|--------|---------|---------|------------|-----------|--------|----------|
| As | Yes | 0.000 | 0.007 | non_normal | wilcoxon | 0.000 | 0.000 |
| Ba | No | 0.538 | 0.000 | non_normal | wilcoxon | 0.114 | 0.538 |
| Cd | Yes | 0.000 | 0.013 | non_normal | wilcoxon | 0.000 | 0.000 |
| Co | Yes | 0.000 | 0.005 | non_normal | wilcoxon | 0.000 | 0.000 |
| Cr | Yes | 0.001 | 0.007 | non_normal | wilcoxon | 0.002 | 0.001 |
| Cu | Yes | 0.000 | 0.000 | non_normal | wilcoxon | 0.003 | 0.000 |
| Fe | No | 0.815 | 0.000 | non_normal | wilcoxon | 0.188 | 0.815 |
| Mn | Yes | 0.000 | 0.001 | non_normal | wilcoxon | 0.000 | 0.000 |
| Mo | Yes | 0.009 | 0.000 | non_normal | wilcoxon | 0.034 | 0.009 |
| Ni | Yes | 0.000 | 0.637 | normal | t test | 0.000 | 0.000 |
| Pb | Yes | 0.001 | 0.000 | non_normal | wilcoxon | 0.469 | 0.001 |
| V | Yes | 0.000 | 0.000 | non_normal | wilcoxon | 0.004 | 0.000 |
| Zn | Yes | 0.000 | 0.019 | non_normal | wilcoxon | 0.000 | 0.000 |



**Table 5:** Are dust corrected plankton concentrations statistically different from zero for 2014 data?

| Element | > zero | p_value | shapiro | not_normal | test_used | t_test | wilcoxon |
|---------|--------|---------|---------|------------|-----------|--------|----------|
| As | Yes | 0.000 | 0.156 | normal | t test | 0.000 | _ |
| Ba | No | 0.979 | 0.019 | non_normal | wilcoxon | _ | 0.979 |
| Cd | Yes | 0.001 | 0.059 | normal | t test | 0.001 | _ |
| Co | Yes | 0.000 | 0.017 | non_normal | wilcoxon | _ | 0.000 |
| Cr | No | 0.102 | 0.022 | non_normal | wilcoxon | _ | 0.102 |
| Cu | Yes | 0.000 | 0.000 | non_normal | wilcoxon | _ | 0.000 |
| Fe | Yes | 0.013 | 0.923 | normal | t test | 0.013 | _ |
| Mn | Yes | 0.046 | 0.000 | non_normal | wilcoxon | _ | 0.046 |
| Mo | No | 0.995 | 0.000 | non_normal | wilcoxon | _ | 0.995 |
| Ni | Yes | 0.003 | 0.817 | normal | t test | 0.003 | _ |
| Pb | Yes | 0.012 | 0.000 | non_normal | wilcoxon | _ | 0.012 |
| V | Yes | 0.000 | 0.017 | non_normal | wilcoxon | _ | 0.000 |
| Zn | Yes | 0.000 | 0.004 | non_normal | wilcoxon | _ | 0.000 |





**Table 6:** Percent of total concentration that is represented by E (the difference between total minus dust contribution)

| Element | % E (2012) | % E (20414) |
| --- | --- | --- |
| Al | 0 | 0 |
| As | 98 | 91 |
| Ba | 29 | -28 |
| Cd | 98 | 87 |
| Co | 48 | 40 |
| Cr | 17 | 11 |
| Cu | 90 | 90 |
| Fe | 3 | 9 |
| Mn | 44 | 49 |
| Mo | 99 | -X |
| Ni | 33 | 31 |
| Pb | 55 | 98 |
| V | 46 | 52 |
| Zn | 85 | 82 |



**Team list**

Dr. Oguz Yigiterhan – Qatar University

Dr. Ebrahim M.A.S Al-Ansari – Qatar University

Ms. Alex Nelson – University of Washington

Dr. Mohamed A.R. Abdel-Moati – Qatar Ministry of Municipality & Environment

Mr. Jesse Turner – University of Washington

Mr. Hamood A. Alsaadi – Qatar University

Mrs. Barbara Paul – University of Washington

Dr. Ibrahim A. Al-Maslamani – Qatar University

Dr. Mehsin A. Al-Ansi Al-Yafei – Qatar University

Dr. James W. Murray – University of Washington

**Author contribution**

*Oguz Yigiterhan:* The first and corresponding author; NPRP Project PI, Contribution for dust and plankton sampling (all campaigns), Data processing for figures and tables, Major contribution for NPRP project & manuscript text writing (Material & Methods; Results, Discussion), Tables 1, 2, 3 & Figures 1, 2, 6, 7.

*Ebrahim M.A.S Al-Ansari:* NPRP Project PI, Sampling cruises Chief Scientist & organizer; coordination of the NPRP project sampling campaigns; Major technical and logistic support for every steps of NPRP project.

*Alex Nelson:* 2014 Plankton sampling, lab processing & acid digestion, Text writing (Introduction).

*Mohamed A.R. Abdel-Moati:* NPRP Project Participant; Contribution for the manuscript text editing. Provided passive dust samplers and obtained required permissions for dust sampling.



*Jesse Turner*: Tables 1, Supplementary materials Table S1, Table S2, Building Figures 3, 4. Data processing for leached concentrations.

*Hamood A. Alsaadi*: All sample digestion, all ICP-OES analysis for plankton and dust samples, raw data processing, QA & QC studies.

*Barbara Paul*: Building Table 6; Figure 5. Cruise participant during 2014 plankton sampling campaigns. Lab work for cruise preparation and after cruise sample handling, filtration and sample acid digestion. Data processing for calculations Excess concentrations. All leaching experiments were done by UW group members under her leadership.

*Ibrahim A. Al-Maslamani:* NPRP Project Co-PI; Plankton sampling, sample gear providing, cruise sampling strategy and logistic support. Designing the experiments. Contributed for plankton sampling.

*Mehsin A. Al-Ansi Al-Yafei:* NPRP Project Lead-PI. Dr. Mehsin took a role for official leader of the project. He has worked for getting required coast guard permissions, provided speedboats, and joined sampling campaigns, helped for writing project and some text in the manuscript.

*James W. Murray:* NPRP Project Co Lead-PI, Senior Author, Major contribution for manuscript text writing (all chapters), editing; Building Tables 4, 5. Dr. Murray has also shown great contribution to this research as being team leader of the UW-Oceanography research team.

**Competing interests**

The authors declare that they have no conflict of interest.

**Disclaimer**

The manuscripts contents are solely the responsibility of the authors and do not necessarily represent the official views of the Qatar National Research Fund.




**Acknowledgement**

The authors would like to thank Qatar National Research Fund (QNRF) for funding and supporting this project (NPRP6-1457-1-272). The dedicated effort exerted by the Captain and

crew of Qatar University R/V Janan during sampling was greatly appreciated. The technical help and support of Mr. Caesar Flonasca Sorino, Mr. Reyniel M. Gasang, Mr. Faisal Muthar Al-Quaiti, Mr. Shafeeq Hamza, Mr. Cherriath A. Cheriath and Mr. Mehmet Demirel from QU-ESC, during sampling, preparation, pretreatment and analysis are acknowledged. Colton Miller (School of Environmental and Forest Sciences, University of Washington) helped with the statistical analysis.

We would like to thank Mr. Jassem Abdulaziz Al-Thani for his valuable comments and edits. We highly appreciate Qatar University – Environmental Sciences Center, Ministry of Municipality and Environment, and University of Washington-School of Oceanography administration and staff for supporting this project at various stages. This study was made possible by a grant from the Qatar National Research Fund (QNRF) under the National Priorities Research Program award number

NPRP 6-1457-1-272. The manuscripts contents are solely the responsibility of the authors and do not necessarily represent the official views of the Qatar National Research Fund.



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
