# Peer review of "Trace Element Composition of Size Fractionated Suspended Particulate Matter Samples from the Qatari Exclusive Economic Zone of the Arabian Gulf; The Role of Atmospheric Dust"

_Biogeosciences, 2019_

## Referee Comment (RC1) · Anonymous Referee #1 · 20 Jun 2019

"The Trace Element Composition of Size Fractionated Suspended Particulate Matter Samples from the Qatari EEZ of the Arabian Gulf: The Role of Atmospheric Dust"

by Yigiterhan et al.,

The paper presents new data set on high precision measurements of trace element concentrations in bulk particulate matter of two size fractions collected by net tow samples from the EEZ of Qatar, Arabian Gulf. The researcher differentiated between lithogenic and biogenic sources of the elements implying correction using dust com-

position. Furthermore, relation between the excess metal concentrations with distance from the coast was used to ascertain the anthropogenic sources. The work carried out is impressive and will significantly improve the knowledge of biogeochemistry of trace elements in this region. Overall the manuscript is clear and easy to follow. However, I suggest minor revision which will further improve the scientific understanding of the study performed as well the quality of the manuscript. The field campaigns carried out during this research is separated by not only years but season. First campaign performed during October, 2012 where as in 2014 samples were collected in April. Referring to Table 2 and 3, we see prominent changes in elemental compositions (both total and excess) particularly in the areas which were revisited (Doha and Dukhan). Such seasonality is not reported or discussed.

Specific comments have been mentioned below: Line 3-5, Page 2 and Line 12-17 Page 7: As stated, researcher didn't manually characterized phytoplankton and zooplankton fractions in their two net-tow samples. It would be wise not to generalize $50\mu$m fraction as phytoplankton and $200\mu$m as zooplankton. Particularly a $50\mu$m net-tow would also capture micro zooplankton. In fact, in tables and figures the author took care about this by stating bulk plankton or small net tow.

Line 6-7, Page 2: The line is misleading. Sampling campaigns were distinctive with varying space and time. 11sites were sampled during 2012 whereas in 2014 six stations were sampled.

Line 30, Page 2: Multiple key words implying same meaning can be removed. E.g., Particulate matter and marine particle, Elemental composition and Trace metal etc.

Line 26, Page 6: Shraawoo's Island

Page 6-7: Please provide depth range among the sampling locations.

Line 20, Page 13: It is mentioned that "Unfortunately, neither Ca nor P analyses were included in this data set.", however, authors presented Ca/Al data from net tow samples

in Fig.6

Line 12, Page 14: dust instead of "duct"

Line 8, Page 15: HAc-HyHCl instead of "HAc:HyHCl"

Line 15, Page 19: Ca is mentioned as biogenic/anthropogenic element but not included in table 6.

Line 10, Page 20: Study occupied entirely in the EEZ of Qatar and doesn't represent entire Arabian Gulf.

Figure Captions:

Figure 1: Figure represents sampling locations during 2012 campaign only.

Figure 2: Near shore sampling were performed during 2014.

Figure 7: Refrain from stating phytoplankton

Missing references:

Turekian 1977

Knauer and Martin, 1981

---

## Referee Comment (RC2) · Anonymous Referee #2 · 15 Jul 2019

The manuscript on The Trace Element Composition of Size Fractionated Suspended Particulate Matter Samples from the Qatari EEZ of the Arabian Gulf: The Role of Atmospheric Dust by Yigiterhan et al presents work on the suspended particulate matter (SPM) from the Qatari EEZ. The samples have been collected during October 2012 and 2014. They have also used dust samples from the land that were previously collected. Trace element composition data of SPM is compared with that of leached, unleached dust, UCC and also applied various corrections like salt lithogenic corrections to get the clear idea of the source of the SPM. They have normalized the data

with Al and also calculated excess metals using atmospheric dust as the background and fate of the dust reaching the EEZ is discussed. With help of the data the authors have distinguished between lithogenic and anthropogenic trace metals reaching the EEZ. The data is of interest as this is the first report and includes systematic study that will help in understanding the biogeochemistry. Abstract, Introduction, Study area, methods, Results discussions and conclusions are clear. Overall, manuscript is nicely written with the clarity that readers will understand. This manuscript has potential and I would suggest that the manuscript may be accepted with moderate revision as I find that there lot of repetitions in the text. The text could be further improved.

Specific comments have been included in the pdf attached.

I would suggest to reduce the number of figures or add them to the supplement

Results and discussion could be combined as same things are repeated.

Please check time of sample collection October or April?

Check tables 2 and 3- same stations during two different years? Change the tables or the captions.

Please also note the supplement to this comment:
https://www.biogeosciences-discuss.net/bg-2019-183/bg-2019-183-RC2-supplement.pdf

**Supplement:**

**The Trace Element Composition of Size Fractionated Suspended Particulate Matter Samples from the Qatari EEZ of the Arabian Gulf: The Role of Atmospheric Dust**

[1*]Oguz Yigiterhan, [1]Ebrahim M.A.S Al-Ansari, [2]Alex Nelson, [3]Mohamed A.R. Abdel-Moati, [2]Jesse Turner, [1]Hamood A. Alsaadi, [2]Barbara Paul, [4]Ibrahim A. Al-Maslamani, [5]Mehsin A. 
[revised manuscript text omitted]

Bulkplankton (50 µm)

| Sample | Location | Dist.(km) | Al | As | | | Ba | | | Cd | | | Co | | | Cr | | | Cu | | |
|---|---|---|---|---|---|---|---|---|---|---|---|---|---|---|---|---|---|---|---|---|---|
| | | | $R$ | $R$ | $L$ | $\underline{E}$ | $R$ | $L$ | $\underline{E}$ | $R$ | $L$ | $\underline{E}$ | $R$ | $L$ | $\underline{E}$ | $R$ | $L$ | $\underline{E}$ | $R$ | $L$ | $\underline{E}$ |
| 22 | Dukhan | 8.77 | 4,129 | 8.40 | 0.41 | 7.99 | 21.9 | 30.2 | -8.29 | 0.80 | 0.03 | 0.76 | 1.77 | 1.06 | 0.71 | 10.3 | 10.2 | 0.08 | 36.0 | 4.85 | 31.1 |
| 23 | Dukhan | 6.78 | 4,035 | 9.62 | 0.40 | 9.22 | 21.7 | 29.5 | -7.88 | 0.79 | 0.03 | 0.76 | 1.68 | 1.04 | 0.63 | 8.8 | 10.1 | -1.27 | 65.2 | 4.74 | 60.5 |
| 24 | Dukhan | 1.31 | 8,407 | 14.45 | 0.83 | 13.61 | 42.2 | 61.5 | -19.32 | 0.19 | 0.07 | 0.12 | 2.62 | 2.17 | 0.45 | 16.7 | 21.1 | -4.32 | 21.7 | 9.87 | 11.8 |
| 25 | Doha | 6.54 | 4,401 | 3.87 | 0.44 | 3.43 | 28.6 | 32.2 | -3.61 | 0.26 | 0.04 | 0.22 | 1.78 | 1.13 | 0.65 | 15.0 | 11.0 | 3.94 | 311.2 | 5.17 | 306.0 |
| 26 | Doha | 2.20 | 3,266 | 2.01 | 0.32 | 1.68 | 12.5 | 23.9 | -11.37 | 0.09 | 0.03 | 0.06 | 1.41 | 0.84 | 0.57 | 8.9 | 8.2 | 0.69 | 44.8 | 3.84 | 41.0 |
| 27 | Doha | 0.06 | 14,493 | 4.54 | 1.44 | 3.11 | 69.5 | 106.1 | -36.58 | 0.10 | 0.12 | -0.02 | 3.86 | 3.74 | 0.12 | 44.0 | 36.3 | 7.70 | 35.9 | 17.02 | 18.9 |
| **Average** | | | **6,455** | **7.15** | | **6.51** | **32.7** | | **-14.51** | **0.37** | | **0.32** | **2.19** | | **0.52** | **17.3** | | **1.14** | **85.8** | | **78.2** |
| 2x St.Dev. | | | 8,675 | 9.16 | | 9.14 | 41.0 | | 24.04 | 0.67 | | 0.70 | 1.83 | | 0.43 | 27.0 | | 8.38 | 222.7 | | 225.8 |

Zooplankton (200 µm)

| Sample | Location | Dist.(km) | Al | As | | | Ba | | | Cd | | | Co | | | Cr | | | Cu | | |
|---|---|---|---|---|---|---|---|---|---|---|---|---|---|---|---|---|---|---|---|---|---|
| | | | $R$ | $R$ | $L$ | $\underline{E}$ | $R$ | $L$ | $\underline{E}$ | $R$ | $L$ | $\underline{E}$ | $R$ | $L$ | $\underline{E}$ | $R$ | $L$ | $\underline{E}$ | $R$ | $L$ | $\underline{
[revised manuscript text omitted]

---

## Author Comment (AC1) · 13 Sep 2019

Response to Comments of the Reviewer # 1

"The Trace Element Composition of Size Fractionated Suspended Particulate Matter Samples from the Qatari EEZ of the Arabian Gulf: The Role of Atmospheric Dust" by Yigiterhan et al.,

The paper presents new data set on high precision measurements of trace element concentrations in bulk particulate matter of two size fractions collected by net tow

samples from the EEZ of Qatar, Arabian Gulf. The researcher differentiated between lithogenic and biogenic sources of the elements implying correction using dust composition. Furthermore, relation between the excess metal concentrations with distance from the coast was used to ascertain the anthropogenic sources. The work carried out is impressive and will significantly improve the knowledge of biogeochemistry of trace elements in this region. Overall, the manuscript is clear and easy to follow. However, I suggest minor revision, which will further improve the scientific understanding of the study, performed as well the quality of the manuscript. The field campaigns carried out during this research is separated by not only years but season. First campaign performed during October 2012 where as in 2014 samples were collected in April. Referring to Table 2 and 3, we see prominent changes in elemental compositions (both total and excess) particularly in the areas, which were revisited (Doha and Dukhan).

Such seasonality is not reported or discussed.

- Response to comment:

We have not specifically focused on temporal and seasonal variations of size fractionated SPM in our manuscript. We have conducted 2 sampling campaigns in October 2012 and April 2014. The second sampling campaign was not the continuation or repetition of the first one. Due to logistic reasons, we were able conduct the 2nd sampling after a while. Additional samples were collected during a third cruise to in October 2014. The data from these samples will be used in a later publication (Yigiterhan et al., in preparation). During the 1st sampling campaign, the size fractionated net-tow samples were collected from off-shore stations (away from the coast and bay areas), we specially focused to catch the influence of the intense anthropogenic impact of oil and gas industry around the islands and deep water rings, heavy industries located along the southeastern coast, offshore hydrocarbon extraction fields etc. Doha and Dukhan offshore stations were also part of the campaign, which were selected to reflect the influence of desalinization plants and oil fields. All samples were collected out of the bays, away from the coast, relatively loaded with less SPM and reflecting

more integrated coverage of the EEZ. However, in 2014 sampling campaign, as you can see from Figure 2; sampling was conducted from semi-closed bay areas for Doha and Dukhan stations, both from the East and West sides of the Qatar Peninsula, reflecting completely different water characteristics, under large anthropogenic effect due to more re-suspended sediments and dust load. The samples were collected along a linear transect inside the Bays and average composition was used for interpreting the data in the manuscript. That is why we have different metal concentrations between 2 years for the same "named" stations (Doha and Dukhan). These differences in concentrations may not point out the temporal variations. Kindly note that we tried to reflect these compositional variations in Figure 6 and 7 for small and large size fractions and for two campaigns with different sampling characteristics. Rather than focusing on temporal and seasonal variations, compositional change of SPM versus distance were targeted for two different size fractions.

Specific comments have been mentioned below: Comment 1: Line 3-5, Page 2 and Line 12-17 Page 7: As stated, researcher didn't manually characterized phytoplankton and zooplankton fractions in their two net-tow samples. It would be wise not to generalize 50_m fraction as phytoplankton and 200_m as zooplankton. Particularly a 50_m net-tow would also capture micro zooplankton. In fact, in tables and figures the author took care about this by stating bulk plankton or small net tow.

- This was corrected. The use of phytoplankton and zooplankton were removed. Now refer only to 50 and 200 mesh as small and large size fractions.

Comment 2: Line 6-7, Page 2: The line is misleading. Sampling campaigns were distinctive with varying space and time. 11sites were sampled during 2012 whereas in 2014 six stations were sampled.

- This line was corrected

Comment 3: Line 30, Page 2: Multiple key words implying same meaning can be removed. E.g., Particulate matter and marine particle, Elemental composition and

Trace metal etc.

- This was fixed, multiple key words were removed

Comment 4: Line 26, Page 6: Shraawoo's Island

- The name of the island was corrected

Comment 5: Page 6-7: Please provide depth range among the sampling locations.

- The water maximum bottom depth range for 2012 sampling stations were varying between 12 to 55 meters depth. However for the 2014 sampling campaign, the sampling depths were varying between 2 to 5 meters in quite shallow bay areas.

Comment 6: Line 20, Page 13: It is mentioned that "Unfortunately, neither Ca nor P analyses were included in this data set.", however, authors presented Ca/Al data from net tow samples in Fig.6

- The confusion was carefully corrected

Comment 7: Line 12, Page 14: dust instead of "duct"

- The misspelling was corrected

Comment 8: Line 8, Page 15: HAc-HyHCl instead of "HAc:HyHCl"

- The formula was corrected

Comment 9: Line 15, Page 19: Ca is mentioned as biogenic/anthropogenic element but not included in table 6.

- We have fixed this. The text was revised and Ca was deleted from the list of elements in Line 19. The Ca concentrations were analyzed in the 3'rd data set (in publication) but has not been included as a separate table into this manuscript to prevent data dump. On the other hand, kindly note that Ca was in the list of elements analyzed for bot leached and unleached data set of Qatari dust samples. This was essential to observe the influence of CaCO3 dissolution in weak acidic conditions.

Comment 10: Line 10, Page 20: Study occupied entirely in the EEZ of Qatar and doesn't represent entire Arabian Gulf.

- The text was revised

Figure Captions:

Figure 1: Figure represents sampling locations during 2012 campaign only. - Corrected

Figure 2: Near shore sampling were performed during 2014. - Corrected

Figure 7: Refrain from stating phytoplankton - Phytoplankton was removed

Missing References:

Turekian 1977 - The missing reference was added

Knauer and Martin, 1981 - The missing reference was added

---

## Author Comment (AC2) · 13 Sep 2019

Response to Comments of the Reviewer # 2

The manuscript on The Trace Element Composition of Size Fractionated Suspended Particulate Matter Samples from the Qatari EEZ of the Arabian Gulf: The Role of Atmospheric Dust by Yigiterhan et al presents work on the suspended particulate matter (SPM) from the Qatari EEZ. The samples have been collected during October 2012 and 2014. They have also used dust samples from the land that were previously collected. Trace element composition data of SPM is compared with that of leached, unleached dust, UCC and also applied various corrections like salt lithogenic corrections to get the clear idea of the source of the SPM. They have normalized the data with Al and also calculated excess metals using atmospheric dust as the background and fate of the dust reaching the EEZ is discussed. With help of the data, the authors have distinguished between lithogenic and anthropogenic trace metals reaching the EEZ. The data is of interest as this is the first report and includes systematic study that will help in understanding the biogeochemistry. Abstract, Introduction, Study area, methods, Results discussions and conclusions are clear. Overall, manuscript is nicely written with the clarity that readers will understand. This manuscript has potential and I would suggest that the manuscript may be accepted with moderate revision as I find that there are a lot of repetitions in the text. The text could be further improved. Specific comments have been included in the pdf attached.

Comment 1: I would suggest to reduce the number of figures or add them to the supplement

- We feel that all Figures are required and made no changes for keeping the integrity and completeness of the manuscript. We are kindly requesting keeping the figures inside the manuscript.

Comment 2: Results and discussion could be combined as same things are repeated.

- We also feel that the best presentation separates Results from Discussion; because of this reason we preferred to keep Results and Discussion separately.

Comment 3: Please check time of sample collection October or April?

- The text has been revised. Months added in to the manuscript text.

Comment 4: Check tables 2 and 3- same stations during two different years? Change the tables or the captions.

- We have done goal oriented research sampling in 2012 and 2014 campaigns and

added metal concentration data in Table 2 and 3. Kindly see the clarification below that was done for the comments of the other reviewer:

"We have not specifically focused on temporal and seasonal variations of size fractionated SPM in our manuscript. We have conducted 2 sampling campaigns in October 2012 and April 2014. The second sampling campaign was not the continuation or repetition of the first one. Due to logistic reasons, we were able conduct the 2nd sampling after a while. Additional samples were collected during a third cruise to in October 2014. The data from these samples will be used in a later publication (Yigiterhan et al., in preparation). During the 1st sampling campaign, the size fractionated net-tow samples were collected from off-shore stations (away from the coast and bay areas), we specially focused to catch the influence of the intense anthropogenic impact of oil and gas industry around the islands and deep water rings, heavy industries located along the southeastern coast, offshore hydrocarbon extraction fields etc. Doha and Dukhan offshore stations were also part of the campaign, which were selected to reflect the influence of desalinization plants and oil fields. All samples were collected out of the bays, away from the coast, relatively loaded with less SPM and reflecting more integrated coverage of the EEZ. However, in 2014 sampling campaign, as you can see from Figure 2; sampling was conducted from semi-closed bay areas for Doha and Dukhan stations, both from the East and West sides of the Qatar Peninsula, reflecting completely different water characteristics, under large anthropogenic effect due to more re-suspended sediments and dust load. The samples were collected along a linear transect inside the Bays and average composition was used for interpreting the data in the manuscript. That is why we have different metal concentrations between 2 years for the same "named" stations (Doha and Dukhan). These differences in concentrations may not point out the temporal variations. We tried to reflect these compositional variations in Figure 6 and 7 for small and large size fractions and for two campaigns with different sampling characteristics. Rather than focusing on temporal and seasonal variations, compositional change of SPM versus distance were targeted for two different size fractions."

Comment 5: Please also note the supplement to this comment: https://www.biogeosciences-discuss.net/bg-2019-183/bg-2019-183-RC2-    supplement.pdf

- We thank the reviewer for the extensive suggestions. We found them very useful to improve the quality of the manuscript significantly. We incorporated most (but not all) of the revisions suggested, paying special attention to removing duplications. For those edits not adopted, we feel that the short phrases are necessary for transition and stating the whole argument.

---

## Referee Report (RR1)

[referee-annotated manuscript omitted]

---

## Author Response (AR2)

**RESPONSE TO COMMENTS OF REVIEWERS**

**BG-2019-183-referee-report-1:**

Minor revision: In the text of revised manuscript the authors revised phytoplankton (50µm) and zooplankton (200µm) to small and large size fraction. The authors are requested to apply same delineation in Figure (40 and Tables (2 and 3) including their captions.

*Response*:

We made the following changes:

- in the legend of the Figure 4:

The word "Phytoplankton" was changed to "Small size fraction (phytoplankton, 50 µm)"

The word "Zooplankton" was changed to "Large size fraction (zooplankton, 200 µm)"

- in the caption of Table 2:

The wording "bulk plankton"  was changed to "small size fraction (phytoplankton)"

The wording  "larger plankton"  was changed to "large size fraction (zooplankton)"

- in the caption of Table 3:

The wording "bulk plankton" was changed to "small size fraction (phytoplankton)"

The wording  "larger fraction of plankton" was changed to "large size fraction (zooplankton)"

**SPECIFIC COMMENTS:**

**- Line 15 Page 6:** Please do not delete the month October

*Response:*

The error was fixed and the month "October" added back into text as recommended.

The corrected sentence is : "The first expedition was conducted on the R/V Janan from 13 to 14 October   2012 and included eleven sampling sites in the Qatar EEZ, east of the Qatari Peninsula (Fig. 1)."

**- Line 16-17 Page 6:** The sentence "Additional samples were collected during a second cruise to these locations in October 2014. The data from these samples will be used in a later publication (Yigiterhan et al., in preparation)." Is again repeated in the end of this section (Line 4-5 Page-7) and one of them can be removed.

*Response:*

The unnecessary repetition in Page 6, Lines 16-17 was deleted, as recommended by Reviewer 1. The repetition in the end of Section 2.2: Sample collection (Page 7, Lines 4-5) was kept and the sentence was restructured.  Thank you for bringing this error to our attention.

**BG-2019-183-referee-report-2:**

The authors have incorporated the corrections suggested and text has been improved a lot and I feel it is suitable for the publication. There are some minor corrections to be incorporated that have been included in the attached file. There is a word to word repetition in the conclusion. These sentences can be re-written.

*Response*: **-** Highlighted text

*Response*: The highlighted text in Page 19  between the Lines 14 to 19 was reworded. A word to word repetition in conclusion was avoided as suggested by the Reviewer 2.

**SPECIFIC COMMENTS:**

**- Page 3  Line21:** Give reference (e.g...Ref)

*Response*: The missing references were added at the end of the sentences, as well as in to the list of references. These are: Lam et al 2015; Ohnemus and Lam 2015;  Ohnemus et al., 2016 Wen-Hsuan Liao et al., 2017.

**- Page 7  Line 4:** using speedboats or ship?

*Response*: Research Vessel Janan was added in to the sentence: "The sampling was conducted using R/V Janan, on a linear transect at seven stations from the exit of Doha channel to the border of EEZ of Qatar."

**- Page 8  Line 16:** Delete

*Response*: The sentence "Samples were digested in strong acid before analyses" was deleted.

**- Page 8  Line 16:** Highlighted text

*Response*: The highlighted text was deleted as recommended in the previous specific comment.

**- Page 10  Line 14:** have Na concentration between 2,000....

*Response*: The missing words were added into the sentence: "Recent marine carbonate deposits and skeletons have Na concentration between 2,000 to 5,000 ppm or 0.2 to 0.5 % (Land and Hoops, 1973; Milliman, 1974) and the average Na in our dust samples was 1.89 %."

**- Page 12  Line 13:** (column R; Table ...)

*Response*:  "Table 2,3" was added into parenthesis after "column R":
"The raw data (column R; Tables 2, 3) for the sea-salt corrected composition of the size fractionated marine particulate matter samples from the 2012 and 2014 cruises are presented by net tow mesh size."

**- Page 12  Line 15:** Table..

*Response*: "Table 2,3" was added into parenthesis after "column L":
"We calculated the lithogenic contributions for each element (column L; Tables 2, 3) using average total (unleached) Qatari Dust (from Table 1)."

**- Page 12  Line 16:** Table..

*Response*: "Table 2,3" was added into parenthesis after "column E":
"and the resulting excess concentrations for each element (column E; Tables 2, 3)."

**- Page 12  Line 21:** Table..

*Response*: Table numbers were given in a parenthesis at the end of the sentence:
"For some elements in some samples, the dust correction (L) was larger than the original signal (R) and this resulted in negative values for excess metal (E) concentrations (Tables 2, 3)."

**- Page 13  Line 8:** (E; Table..)

*Response*: Tables 2 and 3 were referred inside the text.
"We conducted statistical tests to determine if the cumulative means of the excess concentrations (E) in Tables 2 and 3 were different from zero."

**- Page 13  Line 19:** (Table..)

*Response*: Tables 4 and 5 were referred inside the text.
"For the 2012 sample set, only Ni was normally distributed (Table 4). For 2014 As, Cd, Fe and Ni were normal (Table 5)."

**- Page 13  Line 22:** Table..

*Response*: Table numbers 4 and 5 were given in a parenthesis at the end of the sentence:
"Those p values are shown in the 3rd column. The answer to the question "Is the value of column E different from zero?" is given in the 2nd column (Tables 4, 5)."

**- Page 16  Line 24:** the year

*Response*: The missing word "the year" was added into the sentence.
"For the year 2014, Ba, Cr and Mo were not statistically different from zero."

**- Page 17  Line 2:** Vanadium, Symbol/short forms are generally not written in the beginning of sentences.

*Response*: Corrected: "Vanadium increases with Al but at a higher rate."

**- Page17  Line 3:** Barium

*Response*: Corrected: "Barium increases with Al but at a lower rate."

**-Page 17  Line 6:** Year

*Response*: The missing word" year" was added into the sentence:
"Molybdenum mostly increases with Al in agreement with Qatari dust with the exception of six samples, from the year 2012 data set, at low Al that have large excess Mo. Arsenic and Cu are uniformly higher than expected for Qatari dust and Cd is higher, especially at low Al concentrations."

**- Page 19  Line 14:** Highlighted text

*Response*: The highlighted text was reworded as suggested.

**- Page 19  Line 15:** Word to word repeated from discussions. Please reword it

*Response*: A word to word repetition in conclusion was corrected as suggested by Reviewer 2. The text in Page 19  between the Lines 14 to 19 was re-written.

[revised manuscript text omitted]